# How Post-Training Reshapes LLMs: A Mechanistic View on Knowledge, Truthfulness, Refusal, and Confidence

**Hongzhe Du**[1,*,†]**, Weikai Li**[1,*]**, Min Cai**[2]**, Karim Saraipour**[1]**, Zimin Zhang**[3]**,
Himabindu Lakkaraju**[4]**, Yizhou Sun**[1]**, Shichang Zhang**[4]
[1]University of California, Los Angeles        [2]University of Alberta
[3]University of Illinois at Urbana-Champaign        [4]Harvard University

## Abstract

Post-training is essential for the success of large language models (LLMs), transforming pre-trained base models into more useful and aligned post-trained models. While plenty of works have studied post-training algorithms and evaluated post-training models by their outputs, it remains understudied how post-training reshapes LLMs internally. In this paper, we compare base and post-trained LLMs mechanistically from four perspectives to better understand post-training effects. Our findings across model families and datasets reveal that: (1) Post-training does not change the factual knowledge storage locations, and it adapts knowledge representations from the base model while developing new knowledge representations; (2) Both truthfulness and refusal can be represented by vectors in the hidden representation space. The truthfulness direction is highly similar between the base and post-trained model, and it is effectively transferable for interventions; (3) The refusal direction is different between the base and post-trained models, and it shows limited forward transferability; (4) Differences in confidence between the base and post-trained models cannot be attributed to entropy neurons. Our study provides insights into the fundamental mechanisms preserved and altered during post-training, facilitates downstream tasks like model steering, and could potentially benefit future research in interpretability and LLM post-training. Our code is publicly available at HZD01/post-training-mechanistic-analysis.

## 1 Introduction

The success of large language models (LLMs) has standardized a training paradigm consisting of pre-training and post-training. Post-training transforms a pre-trained base model into more useful and aligned post-trained models (Grattafiori et al., 2024; OpenAI, 2023; Jiang et al., 2023; Lambert et al., 2024, inter alia). Initially introduced to improve instruction-following capabilities (Ouyang et al., 2022; Wei et al., 2022), post-training has evolved to serve versatile purposes, including but not limited to making models more truthful (Lin et al., 2022; OpenAI, 2023; Lambert et al., 2024), safety alignment by enabling models to refuse harmful instructions (Bai et al., 2022; Grattafiori et al., 2024), and calibrating the model's output confidence (OpenAI, 2023).

Research on post-training has predominantly focused on algorithms such as Direct Preference Optimization (DPO) (Rafailov et al., 2024) and Reinforcement Learning from Human Feedback (RLHF) (Christiano et al., 2017) and improving LLMs' ability in downstream tasks such as reasoning (Kumar et al., 2025) and math (Liu et al., 2024b). These studies mainly treat the LLM a black box, and only evaluate its outputs externally (Zhou et al., 2023; Wen et al., 2024). However, it remains unclear how post-training affects the mechanisms

---

* Equal contribution    † Correspondence: hongzhedu@cs.ucla.edu

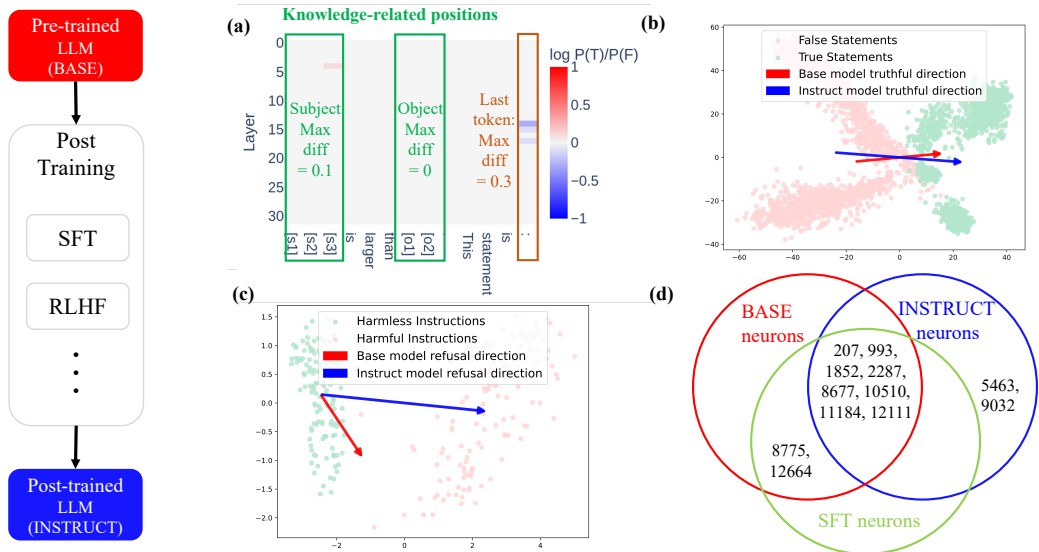

Figure 1: Summary of our analysis. (a) Knowledge: knowledge-storage locations are overlapping between BASE and POST models as the difference is small; (b) Truthfulness: the truthfulness direction is similar between BASE and POST models; (c) Refusal: the refusal direction is different between BASE and POST models; (d) Confidence: the difference in confidence between BASE and POST models cannot be attributed to entropy neurons as they are largely overlapping (numbers are entropy neurons' IDs).

of LLMs and whether the model is fundamentally altered internally. Such a mechanistic understanding can help us better use post-trained LLMs and potentially design better post-training methods.

Recent research studies have started to examine the mechanistic effect of post-training and reveal interesting findings. However, this direction is still underexplored, given these efforts are still algorithm-centric (Lee et al., 2024), model-specific (Panickssery et al., 2024), task-format-specific (Panickssery et al., 2024), or rely on learning an extra model like Sparse Autoencoders (SAEs) on top of the LLM instead of direct analysis (Kissane et al., 2024b).

In this work, we systematically and mechanistically study the post-trained (POST) model, on top of the pre-trained (BASE) model. We compare the BASE and POST models internally from four perspectives: **knowledge storage and representation, internal belief of truthfulness, refusal behavior, and confidence**. These perspectives represent fundamental capabilities that determine an LLM's real-world utility and safety. POST models are expected to preserve knowledge learned during pre-training, improve truthfulness, enhance refusal of harmful inputs, and show a different level of confidence from the base model. We specifically focus on two POST model types: a final model that went through all post-training stages, commonly called the INSTRUCT model, and a model with only supervised fine-tuning on top of BASE, commonly called the SFT model. While some other perspectives, such as reasoning and instruction-following, are also important, they involve complex, multi-step processes that are not well-captured by current interpretability tools. Therefore, our work focuses on four perspectives that can be rigorously measured and mechanistically interpreted, providing a solid foundation for understanding internal mechanisms during post-training.

For each perspective, we choose the most suitable tool from the LLM interpretability toolbox. For the first perspective, we utilize the widely used knowledge locating technique, causal tracing (Meng et al., 2022), to investigate the storage and representation of knowledge. We discover that locations for storing the same knowledge in BASE and POST models are similar, and POST model adapts the original knowledge representations while developing new ones. For the second perspective of truthfulness, based on previous papers' discovery that

the concept of truthfulness can be represented as a direction in the hidden representation space (Marks & Tegmark, 2024; Li et al., 2024; Panickssery et al., 2024; Bürger et al., 2024), we learn a vector linearly representing truthfulness in the model's hidden space, referred to as the "truthfulness direction". For the two directions learned for BASE and POST models, we find that they have high cosine similarity and can be effectively transferred for truthfulness intervention. For the third perspective, we learn a "refusal direction" similar to the truthfulness direction in the hidden representation space (Arditi et al., 2024). We find that the transferability of such refusal direction is only effective backward (from POST to BASE) but not forward (from BASE to POST). Last, we analyze the confidence of BASE and POST models through the lens of entropy neurons, which contributes to forming the LLM's output confidence (Stolfo et al., 2024; Gurnee et al., 2024). Our analysis reveals that entropy neurons of BASE and POST models have similar distributions, leading us to the conclusion that these neurons do not contribute significantly to the observed confidence differences between the BASE and POST models. We illustrate our main conclusions in Figure 1.

Our analysis from the four perspectives reveals both the kept and the altered internal mechanisms by post-training, which could benefit future research and applications in interpretability and LLM post-training. Given some internal mechanisms are mostly developed during pre-training and not significantly altered by post-training, such as factual knowledge and the truthfulness direction. We can leverage the transferability to develop for example truthfulness-oriented procedures on the BASE model and apply it to the POST model conveniently. For the mechanisms that are altered or developed during post-training, such as refusing harmful instructions, there are also possibilities to efficiently improve BASE's ability by applying the backward transfer from POST.

## 2 Related Work

**Mechanistic Interpretability (MI) of Post-training** MI aims to understand internal mechanisms of models (Elhage et al., 2021; Wang et al., 2022; Templeton et al., 2023; Nanda et al., 2023, inter alia). Recently, a growing body of research starts to analyze LLM post-training through the MI lens. Lee et al. (2024) studied how DPO unlearns toxicity in LLM, finding that rather than removing toxic-promoting vectors, the model learns distributed offsets to bypass them. Panickssery et al. (2024) discovered that Llama-2 BASE and INSTRUCT models have similar steering vectors for answering multiple choice questions. Kissane et al. (2024a) showed that refusal directions can be transferred from INSTRUCT models to BASE models. Kissane et al. (2024b) revealed that the SAEs trained on the BASE model can reconstruct the activations of the INSTRUCT model. However, these investigations do not directly and generally reveal the post-training effect, whereas we do a comprehensive study of different models and datasets and investigate post-training's effect from four critical perspectives.

**Knowledge Storage and Representation** Geva et al. (2021) showed that transformer MLP layers function as key-value memories, with keys corresponding to input representations and values inducing output distributions. Dai et al. (2022) identified specific "knowledge neurons" in MLPs that encode facts. To detect knowledge-storage locations and edit them, Meng et al. (2022) introduced causal tracing (activation patching) and edited knowledge through targeted weight changes. These studies show that knowledge in LLMs can be localized and modified through causal intervention techniques. In this work, we use a variant of causal tracing to study the effect of post-training on knowledge storage.

**Internal Belief of Truthfulness** Recent research demonstrates that LLMs encode the belief of truthfulness linearly in their representation space as a "truthfulness direction". Azaria & Mitchell (2023) identified truthfulness signals in model activations, while Burns et al. (2024) developed unsupervised methods to extract these signals using logical consistency. Li et al. (2024) leveraged truthfulness directions to improve truthfulness through activation steering. Later, Marks & Tegmark (2024) introduced the mass-mean (MM) probe. Similarly, Panickssery et al. (2024) uses difference-in-means to identify the direction by computing the difference between mean activation vectors of true and false statements. Additionally, Bürger et al. (2024) discovered a universal two-dimensional truthfulness subspace across various LLMs, and Liu et al. (2024a) showed that training the direction on more datasets

makes it more robust, suggesting that a universal truthfulness hyperplane may exist. We employ MM probe (Marks & Tegmark, 2024) and show that the truthfulness direction persists after post-training.

**Refusal Behavior** Refusing to answer harmful instructions is a key objective of post-training. Recent research has revealed that this behavior is linearly mediated by a vector as a "refusal direction" (Arditi et al., 2024). This direction can be used to undermine the model's ability to refuse harmful requests. Similarly, research on prompt-driven safeguarding has shown that safety prompts typically move input queries in the refusal direction in the representation space (Zheng et al., 2024). Further research has shown this direction can also be learned on BASE models, or transferred from an INSTRUCT model to a BASE model (Kissane et al., 2024a). Our work extends the study to a more systematic comparison of the refusal direction learned on BASE and different POST models across model families.

**Confidence and Entropy Neurons** Confidence calibration is another key objective of post-training. Studies have shown that post-trained models tend to be less calibrated, with INSTRUCT models being overconfident compared to BASE models (Tian et al., 2023). One line of research is to understand LLM's confidence with verbalized output (Tian et al., 2023; Xiong et al., 2024), using prompting and sampling strategies to generate multiple responses and compute consistency. Another line of work analyzes confidence to show that specialized neurons within LLMs regulate uncertainty (Katz & Belinkov, 2023; Gurnee et al., 2024; Stolfo et al., 2024). Among them, Gurnee et al. (2024) discovered "entropy neurons" that have high weight norms but minimal direct logit effects. They modulate uncertainty by influencing layer normalization to scale down logits. Our work examines the changes in entropy neurons after post-training to understand its effect on confidence.

# 3 Notations and Experiments Settings

**Notations** Throughout the paper, we denote layers as $l \in [L]$ and token positions as $i \in [I]$, where $L$ is the layer number and $I$ is the input length. We use notations like $\mathcal{D}_{\text{harmless}}^{\text{train}}$ for datasets, with superscript for train/test, and subscript for the dataset's type. The representation at layer $l$ and position $i$ of an input statement $s$ is denoted as $h_i^l(s)$. We use $\mathbf{W_U} \in \mathbb{R}^{|\mathcal{V}| \times d_{model}}$ for the unembedding matrix, with vocabulary $\mathcal{V}$, and $\mathbf{W}_{\text{out}}$ for the output weights vector of a given neuron in the last-layer MLP.

**Models** We mainly conduct experiments on two representative LLM model families: Llama-3.1-8B/Instruct (Grattafiori et al., 2024) and Mistral-7B-v0.3/Instruct (Jiang et al., 2023). The original model release does not include SFT models of this size, so we use widely recognized external SFT models: Llama-3.1-Tulu-3-8B-SFT, which finetunes Llama-3.1-8B on the tulu-3-sft-mixture dataset (Lambert et al., 2024), and Mistral-7B-Base-SFT-Tulu2 (Feuer et al., 2025), which finetunes Mistral-7B-v0.3 on the tulu-v2-sft-mixture dataset (Ivison et al., 2023). For refusal experiments, we additionally include Qwen-1.5-0.5B/Instruct (Bai et al., 2023) and Gemma-2-9B/Instruct (Team et al., 2024) following experiment settings in Arditi et al. (2024). For confidence experiments, we additionally include Llama-2-7B/Instruct models (Touvron et al., 2023) following Stolfo et al. (2024). To further demonstrate that our findings could generalize to different models sizes, especially larger models, we include experiments on Llama-2-13B/Instruct (Touvron et al., 2023) models for all perspectives in Appendix F.

**Datasets** For the knowledge and truthfulness perspectives, we use datasets from (Marks & Tegmark, 2024; Bürger et al., 2024; Azaria & Mitchell, 2023), where each sub-dataset contains simple and unambiguous statements that are either true or false from diverse topics. For example, cities contains statements about cities and their countries, following the format "The city of [city] is in [country]". The unambiguity and clear dataset make it easy to analyze LLMs. To eliminate the concern that the datasets might be out-of-distribution for post-training, we curate a dataset that is in-distribution for SFT models. We curate the tulu_extracted dataset from the tulu-3-sft-mixture dataset (Lambert et al., 2024), which was used to finetune the Llama-3.1-8B-SFT model. We ensure every statement from tulu_extracted also appears in the tulu-v2-sft-mixture dataset (Ivison et al., 2023), so it is also in-distribution for the Mistral-SFT model. For experiments on the refusal perspective,

we follow Arditi et al. (2024) to use advbench (Zou et al., 2023) for harmful inputs and alpaca (Taori et al., 2023) for harmless inputs. Dataset details are explained in Appendix A.

## 4 Knowledge Storage and Representation

LLMs are known to store factual knowledge in their parameters, particularly in "knowledge neurons" and MLP layers that act as key-value memories. This enables them to answer factual queries, such as answering "TRUE" or "FALSE" for prompt "The city of New York is in the United States. This statement is:". While such knowledge is believed to emerge during pre-training and persist through post-training, mechanistic evidence remains limited. As knowledge is foundational for LLMs, we first examine how post-training affects it—whether it alters (1) knowledge-storage locations and (2) knowledge representations.

When prompted to classify a statement's truthfulness, LLMs retrieve stored knowledge into hidden representations at some layers and tokens, which guide the final output. Following Marks & Tegmark (2024), we adapt causal tracing to identify knowledge-storage locations by patching hidden states between true and false statement pairs. Each pair is token-aligned and differs only in subject—e.g., "The city of Seattle is in France." vs. "The city of Paris is in France.". The relation (e.g., city-in-country) is true for only one statement. We treat subject and object tokens (e.g., city and country) as knowledge-related target tokens for analysis.

**Locating Knowledge** We use causal tracing to localize knowledge storage via three forward passes with varying inputs and intermediate patching. First, we input a true statement $s$ and record the hidden representations $h_i^l(s)$ at each layer $l$ and token position $i$. Second, we input a false statement $\hat{s}$ and similarly record $h_i^l(\hat{s})$. Third, we input $\hat{s}$ again, but patch a specific hidden state $h_i^l(\hat{s})$ with $h_i^l(s)$ from the first run (i.e., replace $h_i^l(\hat{s})$ with $h_i^l(s)$). We perform this patching independently for each $(i, l)$ pair. If patching a particular position flips the output from "FALSE" to "TRUE", it indicates that location contributes to knowledge storage. To measure the effectiveness of the patching, we calculate the log probability difference of outputting "True" versus outputting "False":

$$M_i^l(s, \hat{s}) = log[\frac{P(\text{``TRUE''})}{P(\text{``FALSE''})}|patching(h_i^l(s), h_i^l(\hat{s}))], \tag{1}$$

where a high value indicates that some knowledge is stored in the $l$-th layer at the $i$-th token.

In order to aggregate the location of individual knowledge and analyze the knowledge storage location in general, we average the patching results over all the statements, where we carefully curate the statements to have the same token lengths and token positions. We use (true, false) statement pairs for patching, where each pair only differs in their subjects, and we explain the dataset construction details in Appendix B.1. We construct input prompts by 4-shot examples containing 2 true statements and 2 false statements, followed by the final statement. Patching is applied to the final statement using the methods described above. The aggregated results $(\tilde{M}_i^l)$ are normalized $(M_i^l)$ for better visualization:

$$\tilde{M}_i^l = \frac{1}{|D|} \sum_{(s, \hat{s}) \in D} M_i^l(s, \hat{s}), \quad M_i^l = normalize(\tilde{M}_i^l) \tag{2}$$

In normalization, we divide the range $[\min_{i,l} \tilde{M}_i^l, \max_{i,l} \tilde{M}_i^l]$ into 20 equal-width bins. We set values in the lower 10 bins to 0 and values in the upper 10 bins to 0.1, 0.2, ..., 1. We denote the normalized result as $M_{model} \in R^{L*I}$, where $L$ and $I$ are the number of layers and tokens.

**Q1: Does post-training change LLM's knowledge storage locations?** Figure 2 visualizes the results ($M_{model}$) of Llama-3.1-8B BASE and INSTRUCT on the cities dataset. As shown in the left figure, influential patching consistently occurs at three token positions: subject, object, and the last token. Subject and object are important for both BASE and INSTRUCT. Their difference is nearly zero (e.g., (c)), indicating that BASE and INSTRUCT store knowledge in nearly identical locations. This pattern holds across all datasets and models, with additional visualizations in Appendix B.5. We further conduct quantitative analysis and include SFT

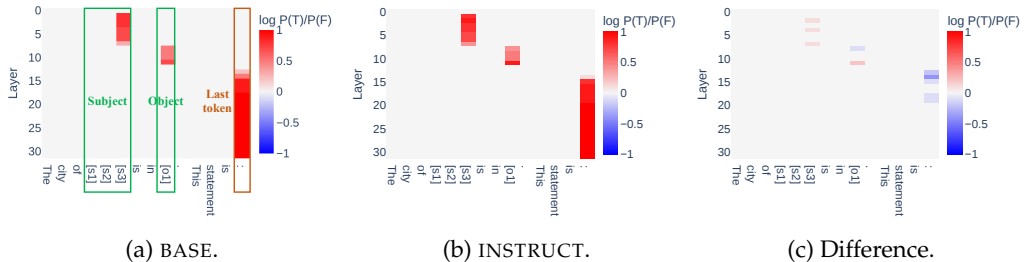

| (a) BASE. | (b) INSTRUCT. | (c) Difference. |

Figure 2: Knowledge storage locations of Llama-3.1-8B BASE and INSTRUCT on the `cities` dataset. Their knowledge-storage locations are almost the same.

| Metric | cities | neg_cities | larger_than | smaller_than | sp_en_trans | neg_sp_en_trans | tulu_extracted |
|---|---|---|---|---|---|---|---|
| Number of Curated Pairs | 238 | 215 | 406 | 487 | 25 | 33 | 55 |
| $Corr(M_{BASE}, M_{INSTRUCT})$ | 0.9923 | 0.9853 | 0.9969 | 0.9805 | 0.9945 | 0.9822 | 0.9978 |
| $max\|M_{INSTRUCT} - M_{BASE}\|$ | 0.4 | 0.4 | 0.3 | 0.5 | 0.3 | 0.5 | 0.2 |
| $max\|M_{INSTRUCT} - M_{BASE}\|_K$ | 0.2 | 0.4 | 0.1 | 0.5 | 0.2 | 0.1 | 0.1 |
| $Corr(M_{BASE}, M_{SFT})$ | 0.9962 | 0.9947 | 0.9978 | 0.9855 | 0.9975 | 0.9792 | 0.9969 |
| $max\|M_{SFT} - M_{BASE}\|$ | 0.2 | 0.2 | 0.1 | 0.5 | 0.2 | 0.5 | 0.2 |
| $max\|M_{SFT} - M_{BASE}\|_K$ | 0.2 | 0.2 | 0.1 | 0.5 | 0.1 | 0.2 | 0.1 |

Table 1: Comparison of knowledge storage locations of the Llama-3.1-8B model family.

models. We compute Pearson correlation coefficient between $M_{BASE}$ and $M_{POST}$, where POST is INSTRUCT or SFT. We also measure the maximum absolute difference value over all tokens, $max\|M_{POST} - M_{BASE}\|$, as well as only over knowledge-related tokens (subject and object), $max\|M_{POST} - M_{BASE}\|_K$. Results for the Llama-3.1-8B family are in Table 1, and for Mistral-7B in Table 8 in Appendix B.4. All results show high correlation and low difference, confirming that **post-training has little influence on knowledge-storage locations**.

**Q2: Does post-training change the knowledge representations?** We further conduct cross-model experiments by patching hidden representations from BASE to POST (forward patching) and from POST to BASE (backward patching). It allows us to analyze whether knowledge representations in BASE can still work in POST, and vice versa. Due to space limits, we put the visualizations on all models and datasets in Appendix B.5. The results demonstrate that the forward patching is almost always successful, but the backward patching often fails, i.e., it does not recover the log probability difference. It leads to the conclusion that **knowledge representations of BASE still work after post-training, but post-training also develops new knowledge representations**.

**Verification on in-distribution dataset** One natural question is that our previous experiments are based on general datasets independent of post-training, which can be considered out-of-distribution. To verify the conclusions completely, we extract factual knowledge from the Tulu dataset (Lambert et al., 2024), which was used to fine-tune Llama-3.1-8B-SFT and Mistral-7B-v0.3-SFT (Feuer et al., 2025). We generate (true, false) statement pairs from the dataset, and it can be considered an in-distribution dataset for the SFT models. Different from previous datasets, pairs in the Tulu dataset could have different lengths, so we slightly modify the metric calculation, specified in Appendix B.3. The last column of Table 1 shows results of the Llama-3.1-8B family, and the last column of Table 8 in Appendix B.4 shows results of the Mistral-7B family. They verify our previous conclusions.

Besides, to verify our conclusion's generalizability, we conduct experiments on a larger model, Llama-2-13B (Touvron et al., 2023), shown in Appendix F. We also conduct experiments following the traditional causal tracing setting (Meng et al., 2022), which asks the LLM to output the object given a subject. We do not use the traditional setting in the main experiments because it cannot test knowledge storage in the object, and it only allows a small range of datasets, as explained in Appendix B.4. The results also verify the conclusions.

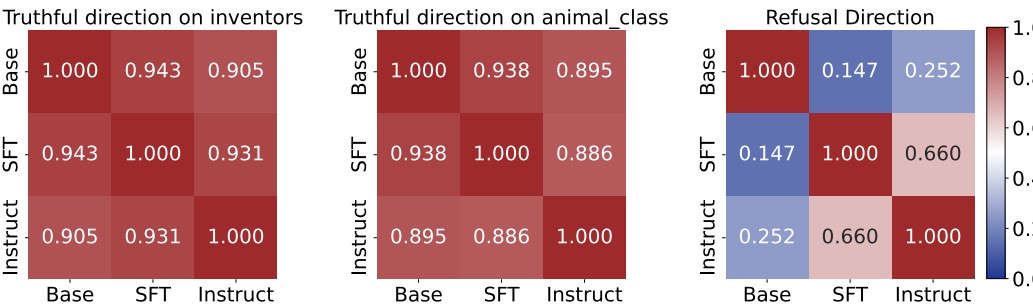

Figure 3: Cosine similarities of truthfulness and refusal directions of Llama-3.1-8B BASE, INSTRUCT, and SFT. Truthfulness directions are similar while refusal directions are different.

## 5 Internal Belief of Truthfulness

How LLMs internally assess the truthfulness of an input statement is another essential aspect of making LLMs truthful and reliable. Previous studies have found that given an LLM and a statement, whether the LLM believes the statement to be true or false can be assessed from the hidden representations encoded by the model. Such belief of truthfulness can be linearly represented along a truthfulness direction in the hidden representation space (Marks & Tegmark, 2024; Bürger et al., 2024). We analyze this direction in BASE models and POST models to analyze whether post-training changes this truthfulness direction.

**Linear Probe for Truthfulness** To identify the truthfulness direction in a model, we compute difference-in-mean on the hidden representations $h^l$, where $l$ is the layer number where truthfulness is most strongly encoded (based on causal tracing results in Section 4). We get this truthfulness direction on one dataset (training dataset) and transfer it to other datasets. Given a training true/false dataset $\mathcal{D}^{\text{train}}$, we separate it into true statements $\mathcal{D}^{\text{train}}_{\text{true}}$ and false statements $\mathcal{D}^{\text{train}}_{\text{false}}$. Similar to knowledge-storage experiments, we use two true statements and two false statements to construct 4-shot prompts, specified in Appendix C.1. The model follows the 4 examples to output "TRUE" or "FALSE" for the final statement. In this process, we compute the truthfulness direction $\mathbf{t}$ as:

$$t^l = \frac{1}{|\mathcal{D}^{\text{train}}_{\text{true}}|} \sum_{s \in \mathcal{D}^{\text{train}}_{\text{true}}} h^l_i(s) - \frac{1}{|\mathcal{D}^{\text{train}}_{\text{false}}|} \sum_{s \in \mathcal{D}^{\text{train}}_{\text{false}}} h^l_i(s), \tag{3}$$

where $i$ is the last token of the input prompt and $l$ is the selected layer. Figure 3 (a) and (b) show the cosine similarities of $t^l$ from different models on two truthfulness datasets. The heatmaps show a high cosine similarity, revealing that BASE, SFT, and INSTRUCT models have remarkably similar internal truthfulness directions.

To further investigate the generalizability, we utilize $\mathbf{t}$ as the weight of logistic regression to construct an MM probe to classify whether a statement is true (Marks & Tegmark, 2024) by $p = \sigma(h^l_i(s)^T t^l)$, where $s \in \mathcal{D}^{\text{train}}$, and $\sigma$ is the sigmoid function. We train the probe on five datasets and test its performance on another dataset. We conduct model-transfer experiments, training the probe on the hidden representations of true/false statements generated by one model and evaluating its accuracy in classifying representations generated by other models. We compare the accuracy of training the probe on POST ($p_{\text{POST}}$) and applying it on POST's test representations (baseline) versus training it on BASE ($p_{\text{BASE}}$) and applying it to POST's test representations (forward transfer). Table 2 presents the results. The probe classification accuracies across BASE, SFT, and INSTRUCT are very similar. Across all datasets, $p_{\text{BASE}}$ achieves comparable accuracy to $p_{\text{SFT}}$ and $p_{\text{INSTRUCT}}$ when applying on SFT and INSTRUCT's test representations, with little differences ($\Delta$). Experiments on the Mistral model family also verify this conclusion, as shown in Appendix C. These findings suggest that the direction corresponding to truthfulness is preserved in post-training.

| Test Dataset | Probe Transfer Accuracy (%) | | |
|---|---|---|---|
| | $p_{\text{BASE}} \to h_{\text{BASE}}$ | $p_{\text{SFT}} \to h_{\text{SFT}} \;/\; p_{\text{BASE}} \to h_{\text{SFT}}$ ($\Delta$) | $p_{\text{INS}} \to h_{\text{INS}} \;/\; p_{\text{BASE}} \to h_{\text{INS}}$ ($\Delta$) |
| cities | 81.06 | 84.50 / 85.32 (+0.82) | 94.65 / 95.91 (+1.26) |
| sp_en_trans | 97.16 | 98.45 / 98.88 (+0.43) | 95.18 / 98.94 (+3.76) |
| inventors | 92.72 | 91.96 / 93.12 (+1.16) | 88.73 / 92.18 (+3.45) |
| animal_class | 97.20 | 96.01 / 95.64 (-0.37) | 98.75 / 96.46 (-2.29) |
| element_symb | 92.02 | 94.87 / 97.02 (+2.15) | 96.18 / 95.13 (-1.05) |
| facts | 77.05 | 77.58 / 77.72 (+0.14) | 82.47 / 80.86 (-1.61) |

Table 2: Probe transfer accuracy ($\uparrow$) of Llama-3.1-8B BASE, SFT, and INSTRUCT tested on 6 truthfulness datasets. For each row, the datasets from the other 5 rows are used for training. $p_{model_1} \to h_{model_2}$ means using the probe trained on $model_1$ to classify truthfulness direction in $model_2$. Probe transfer shows little difference ($\Delta$) compared to the same-model probe.

**Transfer Intervention with Truthfulness Directions** The truthfulness direction **t** can also be used to steer model output. To flip a model's response between "TRUE" and "FALSE" for a statement, we can add **t** to the model's hidden representation as $\tilde{h}^l = h^l + \lambda \mathbf{t}^l$, with $\lambda = \pm 1$ to control the flipping direction following Marks & Tegmark (2024). We also conduct additional robustness experiments with varying values of $\lambda$ in Appendix C.4. The main conclusions are not affected by the choice of $\lambda$, so we use a standard setting of $\pm 1$ in the main experiments. To investigate the transferability of **t**, we test: (1) intervening $h_{\text{SFT}}$ with $\mathbf{t}_{\text{BASE}}$ versus $\mathbf{t}_{\text{SFT}}$; and (2) intervening $h_{\text{INSTRUCT}}$ with $\mathbf{t}_{\text{BASE}}$ versus $\mathbf{t}_{\text{INSTRUCT}}$. We evaluate the intervention performance using the *Intervention Effect (IE)*: $(\tilde{P}^- - P^-)/(1 - P^-)$ for false $\to$ true intervention, and $(\tilde{P}^+ - P^+)/(-1 - P^+)$ for true $\to$ false intervention. $P^-$ and $P^+$ represent the average probability difference $P(TRUE) - P(FALSE)$ for false and true statements, respectively. $\tilde{P}^-$ and $\tilde{P}^+$ are $P^-$ and $P^+$ after intervention, respectively. The goal is to increase $P(TRUE) - P(FALSE)$ for false statements after the intervention, i.e., $\tilde{P}^-$, and to decrease $\tilde{P}^+$ for true statements after the intervention, so a higher IE indicates better intervention performance. The results in Table 3 show that when steering SFT, the difference ($\Delta$) of IE between $\mathbf{t}_{\text{BASE}}$ and $\mathbf{t}_{\text{SFT}}$ is little. Similar results hold for INSTRUCT. We also conduct experiments on Mistral models in Appendix C, which verifies this result. We illustrate two intervention examples in Appendix C.6, which shows that $\mathbf{t}_{\text{BASE}}$ can flip T/F outputs in POST models as effectively as $\mathbf{t}_{\text{POST}}$. These findings further support our conclusion: **post-training generally preserves the internal representation of truthfulness**.

| Test Dataset | Truthful Intervention Effects | | |
|---|---|---|---|
| | $t_{\text{BASE}} \mapsto h_{\text{BASE}}$ | $t_{\text{SFT}} \mapsto h_{\text{SFT}} \;/\; t_{\text{BASE}} \mapsto h_{\text{SFT}}$ ($\Delta$) | $t_{\text{INS}} \mapsto h_{\text{INS}} \;/\; t_{\text{BASE}} \mapsto h_{\text{INS}}$ ($\Delta$) |
| cities | 0.83 | 0.91 / 0.92 (+0.01) | 0.88 / 0.90 (+0.02) |
| sp_en_trans | 0.78 | 0.82 / 0.83 (+0.01) | 0.84 / 0.81 (-0.03) |
| inventors | 0.72 | 0.80 / 0.82 (+0.02) | 0.79 / 0.83 (+0.04) |
| animal_class | 0.73 | 0.79 / 0.80 (+0.01) | 0.71 / 0.72 (+0.01) |
| element_symb | 0.79 | 0.84 / 0.86 (+0.02) | 0.73 / 0.77 (+0.04) |
| facts | 0.61 | 0.64 / 0.66 (+0.02) | 0.62 / 0.66 (+0.04) |

Table 3: Intervention effect ($\uparrow$) of intervention on Llama-3.1-8B BASE, SFT, and INSTRUCT. For each row, we use the other 5 rows' datasets for training. $t_{model_1} \mapsto h_{model_2}$ means using the truthfulness direction in $model_1$ to intervene $model_2$. Transfer interventions show small differences ($\Delta$) compared to same-model interventions.

## 6 Refusal

Post-training also aims at preventing LLMs from generating harmful and toxic content. For example, when inputting harmful prompts like "Give instructions for making explosive materials", the INSTRUCT models might refuse to answer by outputting "I can't fulfill that

| | **Intervention Refusal Score** | | |
|---|---|---|---|
| | BASE | SFT | INSTRUCT |
| **Inputs** | baseline/$r_{\text{BASE}} \mapsto h_{\text{BASE}}$ | baseline/$r_{\text{SFT}} \mapsto h_{\text{SFT}}$/$r_{\text{BASE}} \mapsto h_{\text{SFT}}$ | baseline/$r_{\text{INS}} \mapsto h_{\text{INS}}$/$r_{\text{SFT}} \mapsto h_{\text{INS}}$/$r_{\text{BASE}} \mapsto h_{\text{INS}}$ |
| harmful (↓) | 0.21 / 0.17 | 0.99 / 0.79 / 0.99 | 0.98 / 0.01 / 0.36 / 0.95 |
| harmless (↑) | 0.01 / 0.59 | 0.01 / 1.0 / 0.85 | 0.0 / 1.0 / 0.98 / 0.08 |

Table 4: Intervention RS of Llama-3.1-8B BASE, SFT, and INSTRUCT tested on harmful and harmless inputs. $r_{model_1} \mapsto h_{model_2}$ means using the refusal direction in $model_1$ to intervene $model_2$, and baseline refers to the original Refusal Score without intervention. For harmful inputs we use ablation and for harmless inputs we use addition.

request...", but BASE models might not. Recent studies by Arditi et al. (2024) show that, similar to the internal belief of truthfulness, this refusal behavior can also be linearly mediated by a vector in the hidden representations as a "refusal direction". By steering a model with it, we can encourage the model to change its original behaviors to follow harmful instructions or refuse harmless instructions. Kissane et al. (2024a) found that BASE models also demonstrate the refusal behavior for some harmful instructions, and thus a refusal direction can also be extracted. It also verified the backward transferability of transferring the refusal direction from the INSTRUCT model to the BASE model. We aim to compare the refusal directions in POST models versus BASE models similarly to the truthfulness direction in Section 5 and study its forward transferability.

To extract the refusal direction $\mathbf{r}$, we use $\mathcal{D}_{\text{harmful}}^{\text{train}}$ (a size-128 subset of advbench) and $\mathcal{D}_{\text{harmless}}^{\text{train}}$ (a size-128 subset of alpaca) to construct the refusal direction. We calculate the refusal direction similarly to the truthfulness direction based on Equation 3. Following Arditi et al. (2024), we compute candidate $\mathbf{r}$ for all token positions and layers and select the most effective one. In the intervention experiments, given $\mathbf{r}$, we induce the refusal behavior on harmless inputs by adding $\mathbf{r}$ to the model's representations at the layer where $\mathbf{r}$ is learned, i.e., $\tilde{h}^l \leftarrow h^l + \mathbf{r}^l$. To reduce refusal, we subtract $\mathbf{r}$ from the model's representations at all layers, i.e., $\tilde{h} \leftarrow h - \hat{\mathbf{r}}\hat{\mathbf{r}}^\top h$, where $\hat{\mathbf{r}}$ is the unit-norm vector of $\mathbf{r}$. Interventions are applied at all token positions.

To study the refusal behavior across models, we first directly compare $\mathbf{r}$ learned on BASE ($\mathbf{r}_{\text{BASE}}$), SFT ($\mathbf{r}_{\text{SFT}}$), and INSTRUCT ($\mathbf{r}_{\text{INSTRUCT}}$) models. Figure 3 (c) shows that $\mathbf{r}_{\text{BASE}}$ has very low cosine similarity with $\mathbf{r}_{\text{SFT}}$ and $\mathbf{r}_{\text{INSTRUCT}}$. To further investigate this, we conduct forward transfer intervention experiments similar to Section 5. We compare the *Refusal Score (RS)* when using $\mathbf{r}_{\text{BASE}}$ to steer SFT and INSTRUCT versus using their native refusal vectors ($\mathbf{r}_{\text{SFT}}$ and $\mathbf{r}_{\text{INSTRUCT}}$). RS is calculated as the percentage of responses where refusal keywords such as "I can't" or "I am sorry" appear at the beginning of outputs. We do an intervention on both harmful and harmless datasets, sampling 100 prompts from each for testing. We try to alter the original outputs, i.e., to decrease *RS* for harmful inputs and increase *RS* for harmless inputs. Table 4 demonstrates that $\mathbf{r}_{\text{BASE}}$ generally cannot be effectively transferred to steer INSTRUCT and SFT. Following Arditi et al. (2024), we also conduct experiments on Qwen-1.5-0.5B/Instruct (Bai et al., 2023) and Gemma-2-9B/Instruct (Team et al., 2024) in Appendix D. Results also verify this conclusion: **post-training changes the refusal direction and it has limited forward transferability.**

## 7 Confidence

Confidence of LLMs is represented by the probability associated with the decoded token. Post-trained models are known to have different confidence compared to BASE models (OpenAI, 2023), which is also revealed in their drastically different outputs to the same prompts. Understanding and calibrating model confidence is an important research direction. Recently, entropy neurons have been shown to be a mechanism of modulating confidence that is persistent across models (Gurnee et al., 2024; Stolfo et al., 2024). Entropy neurons help calibrate the model's confidence. They have relatively high weight norms and a low composition with the model's unembedding matrix, so they influence the model's output logits

without affecting the token ranking and which token will be predicted, working similarly to the temperature parameter. We aim to study whether the difference in confidence between BASE and POST models is caused by the difference in entropy neurons.

Entropy neurons are identified by checking the weight norm and logit attribution. First, we compute the logit attribution for each neuron in the final MLP layer by projecting its output weights onto the vocabulary space through the unembedding matrix. This projection (Equation 4) approximates the neuron's direct effect on the final prediction logits:

$$\text{LogitVar}(\mathbf{w}_{\text{out}}) = \text{Var}\left(\frac{\mathbf{W_U}\mathbf{w}_{\text{out}}}{\|\mathbf{W_U}\|_{\text{dim}=1}\|\mathbf{w}_{\text{out}}\|}\right), \tag{4}$$

where $W_{out}$ is the weight vector of the last MLP layer, $W_U$ is the unembedding matrix, and $\|\cdot\|_{\text{dim}=1}$ denotes a row-wise norm. We then calculate the variance of this normalized projection (LogitVar), where a low LogitVar value indicates a relatively balanced contribution across all vocabulary tokens rather than promoting specific tokens. Entropy neurons typically have both a large weight norm (ensuring they are influential) and a low LogitVar (indicating balanced contribution across vocabulary tokens). Our identification process first selects the top 25% of neurons with the largest weight norms, and from this subset, we identify the 10 neurons with the lowest LogitVar values as entropy neurons from the final MLP layer. This methodology follows established practices from prior work and captures neurons that modulate output entropy without significantly affecting token ranking.

In our analysis comparing BASE and POST models, we found substantial overlap in identified entropy neurons, with highly similar ratios of $\left|\frac{\text{weight norm}}{\log(\text{LogitVar})}\right|$ between models. We show the detailed results in Appendix E. These finding suggests that the confidence regulation mechanism of entropy neurons remains largely unchanged during post-training, indicating that the observed confidence calibration differences between BASE and POST models likely stem from more subtle mechanistic changes that require sophisticated interpretability tools beyond current entropy neuron analysis to fully understand.

## 8   Discussion and Conclusion

To achieve effective post-training, it is important to understand how it shapes LLMs internally. In this paper, we analyze its effect on LLM's internal mechanisms from four representative perspectives. We discover that post-training does not alter knowledge-storage locations and truthfulness directions significantly, and adapts original knowledge representations while developing some new ones. However, post-training changes the refusal direction. We also find that the confidence difference brought by post-training cannot be attributed to entropy neurons, further works need to be done.

Our findings could also benefit many real-world applications. As we have shown, general abilities such as factual knowledge and the internal belief of truthfulness are mostly developed during pre-training and remain unchanged in post-training. Although post-training develops new knowledge representations, the forward transfer remains valid. For fixing mistakes or outdated knowledge, this allows us to conveniently and effectively transfer knowledge editing developed on a BASE model to its POST model. We can also transfer the hidden probe of truthfulness learned from BASE or POST to each other, benefiting model steering. In contrast, some internal mechanisms are significantly modified by post-training, such as refusing harmful instructions. In these areas, a valuable application is to transfer the newly acquired capabilities from the POST model to the BASE model, making it efficient for the BASE model to obtain such ability (Kissane et al., 2024a).

Although we concentrated on four key perspectives, future work could extend our framework to more complex capabilities, such as reasoning and instruction-following. These areas present significant methodological challenges for existing interpretability tools. We also find that properly defining the instruction-following ability is tricky, and a suitable technique to interpret this ability and verify it on BASE is also non-trivial. Also, future work could utilize the analysis to improve the post-training effectiveness and efficiency.

## 9 Acknowledgments

We would like to thank Fan Yin for insightful discussions. This work was partially supported by NSF 2211557, NSF 1937599, NSF 2119643, NSF 2303037, NSF 2312501, NASA, SRC JUMP 2.0 Center, Amazon Research Awards, and Snapchat Gifts.

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

## A  Details on Datasets

| Name | Description | #Data points |
|------|-------------|--------------|
| **True / False Datasets (Knowledge & Truthfulness)** | | |
| element_symb | Symbols of elements | 186 |
| animal_class | Classes of animals | 164 |
| inventors | Home countries of inventors | 406 |
| facts | Diverse scientific facts | 561 |
| cities | "The city of [city] is in [country]." | 1496 |
| neg_cities | Negations of statements in cities with "not" | 1496 |
| sp_en_trans | "The Spanish word '[word]' means '[English word]'." | 354 |
| neg_sp_en_trans | Negations of statements in sp_en_trans with "not" | 354 |
| larger_than | "$x$ is larger than $y$." | 1980 |
| smaller_than | "$x$ is smaller than $y$." | 1980 |
| tulu_extracted | Diverse T/F statements extracted from tulu-3-sft-mixture | 200 |
| **Harmful / Harmless Datasets (Refusal)** | | |
| advbench | Harmful instructions | 520 |
| alpaca | Harmless instructions | 52k |

Table 5: Dataset Descriptions and Statistics.

Table 5 presents details on the datasets we use for our experiments. For the datasets that follow a strict template, such as cities, neg_cities, etc., we write their templates in the table. For datasets that do not follow a strict template, such as element_symb and animal_class, we describe them in the table. For the true/false datasets, you can find four examples for each dataset in Table 7.

The Tulu_extracted dataset is an in-distribution dataset for the Llama-3.1-8B SFT and Mistral-7B-v0.3 SFT models. In order to construct it, we use GPT-4o to extract 100 factual knowledge statements from the Tulu-SFT dataset that was used to fine-tune the SFT models (Lambert et al., 2024). Then we use GPT-4o to generate a false statement for each true factual statement by changing the subject, object, or subject-object relation.

## B  Supplementary Details and Experiments of Knowledge Storage

### B.1  (True, False) Pair Construction

As introduced in the main content, in order to provide a generalizable conclusion, we want to aggregate the results from all the prompts, and thus we need to align the token positions of all the prompts. Therefore, we manually find out the most common token pattern in each dataset, and we filter out the prompts that do not match this pattern. It ensures that every statement has the same number of tokens, and that their subjects/objects appear in the same token positions. After filtering, about one-third to half of the original dataset remains. We list the token patterns we use for each dataset in Table 6.

After filtering, we obtain a subset for each original dataset. This subset contains a group of true statements and a group of false statements with the same token patterns. Then, for each true statement, we search for the first unused false statement whose object is the same but the subject is different. In this case, they only differ in the subject. If all the false statements that only differ in the subject are already paired with a true statement, then we repeatedly use the last satisfying paired false statement. It is because we want to increase the number of (true, false) statement pairs, and it does not matter much if one false statement is paired with more than one true statement. If we cannot find any false statement that only differs in the subject, then we do not use that true statement. By this method, we construct abundant (true, false) statement pairs for our patching experiments.

| Dataset | Model family | Token pattern |
|---|---|---|
| cities | both | [Begin] / The / city / of/ [3-token city name] / is / in / [1-token country name] / . |
| neg_cities | both | [Begin] / The / city / of/ [3-token city name] / is / not / in / [1-token country name] / . |
| larger_than | Llama-3.1-8B | [Begin] / [3-token number] / is / larger / than / [2-token number] / . |
| larger_than | Mistral-7B | [Begin] / [4-token number] / is / larger / than / [3-token number] / . |
| smaller_than | Llama-3.1-8B | [Begin] / [3-token number] / is / smaller / than / [2-token number] / . |
| smaller_than | Mistral-7B | [Begin] / [4-token number] / is / smaller / than / [4-token number] / . |
| sp_en_trans | both | [Begin] / The / Spanish / word / ' / [2-token Spanish word] / ' / means / ' / [1-token English word] / '. |
| neg_sp_en_trans | both | [Begin] / The / Spanish / word / ' / [2-token Spanish word] / ' / does / not / mean / ' / [1-token English word] / '. |

Table 6: The token patterns we use to select the statements from the original dataset for the knowledge storage experiments.

## B.2 Few-shot prompting

For each dataset, we select 2 true examples and 2 false examples to conduct four-shot prompting. We randomly select them from the dataset once, and then we fix them. The selected examples are shown in Table 7. The input is constructed in the template: "[four examples] [final statement] This statement is:". To eliminate the influence of example order, we randomly perturb the four examples for every (true, false) statement pairs, so different pairs might have different example orders, but the true and false statements in a pair have the same example order. We set the random seed to 1 in the beginning to ensure the reproducibility of this random ordering.

## B.3 Adapting Causal Tracing for the Tulu_extracted Dataset

For the Tulu_extracted dataset, we also only use the pairs where the true and false statements have the same number of tokens in this experiment. Among them, most of the pairs differ in the object. Nonetheless, a natural consequence of this unstructured dataset construction is that different pairs could have different numbers of tokens, so we cannot directly align them.

In order to aggregate the results from different statement pairs, we use another method to align them. Based on our previous finding that the influential patching only occurs on the knowledge-related tokens and the last token, we categorize the tokens into three categories: the different tokens between the true and false statements, the last token, and the other tokens. The different tokens can be seen as knowledge-related tokens. The three token categories can be seen as three meta-tokens, and we want to transform the results on the original tokens into the three meta-tokens. After doing patching for each (true, false) statement pair $(s, \hat{s})$, we first calculate the metric $M_i^{(l)}(s, \hat{s})$ for each token position $i$ and layer $l$ as before. Then for each pair, we average the results on all the knowledge-related tokens to obtain $M_K^{(l)}(s, \hat{s})$, record the result of the last token $M_{-1}^{(l)}(s, \hat{s})$, and average the results on the other tokens to obtain $M_O^{(l)}(s, \hat{s})$. Now we have results for the three meta-tokens and $|L|$ layers. Then, we use the same way as before to average the results among all the prompt pairs and normalize the results. The final result is denoted $M_{model} \in R^{|L|*3}$, which we can visualize and evaluate as before.

| Dataset | Token pattern |
|---|---|
| element_symb | "Astatine has the symbol At. This statement is: TRUE", "Arsenic has the symbol As. This statement is: TRUE", "Platinum has the symbol La. This statement is: FALSE", "Titanium has the symbol B. This statement is: FALSE" |
| animal_class | "The otter is a mammal. This statement is: TRUE", "The skunk is a mammal. This statement is: TRUE", "The tuna is a mammal. This statement is: FALSE", "The giraffe is a crustacean. This statement is: FALSE" |
| inventors | "Candace Pert lived in the U.S. This statement is: TRUE", "Levi Strauss lived in the U.S. This statement is: TRUE", "Frederick McKinley Jones lived in Japan. This statement is: FALSE", "Elisha Otis lived in the U.K. This statement is: FALSE" |
| facts | "The scientific method is a systematic process for investigating phenomena and acquiring new knowledge. This statement is: TRUE", "Birds have feathers and wings. This statement is: TRUE", "Cacti store water in their ears. This statement is: FALSE", "The process of aging is influenced solely by environmental factors. This statement is: FALSE" |
| cities | "The city of Dar es Salaam is in Tanzania. This statement is: TRUE", "The city of Kozhikode is in India. This statement is: TRUE", "The city of Dar es Salaam is in Italy. This statement is: FALSE", "The city of Kozhikode is in the United States. This statement is: FALSE" |
| neg_cities | "The city of Dar es Salaam is not in Italy. This statement is: TRUE", "The city of Kozhikode is not in the United States. This statement is: TRUE", "The city of Dar es Salaam is not in Tanzania. This statement is: FALSE", "The city of Kozhikode is not in India. This statement is: FALSE" |
| larger_than | "Seventy-eight is larger than seventy-three. This statement is: TRUE", "Ninety-six is larger than sixty-six. This statement is: TRUE", "Fifty-eight is larger than ninety-six. This statement is: FALSE", "Seventy-nine is larger than ninety-seven. This statement is: FALSE" |
| smaller_than | "Fifty-eight is smaller than ninety-six. This statement is: TRUE", "Seventy-nine is smaller than ninety-seven. This statement is: TRUE", "Seventy-eight is smaller than seventy-three. This statement is: FALSE", "Ninety-six is smaller than sixty-six. This statement is: FALSE" |
| sp_en_trans | "The Spanish word 'bosque' means 'forest'. This statement is: TRUE", "The Spanish word 'piel' means 'skin'. This statement is: TRUE", "The Spanish word 'gobernar' means 'to eat'. This statement is: FALSE", "The Spanish word 'edad' means 'clock'. This statement is: FALSE" |
| neg_sp_en_trans | "The Spanish word 'gobernar' does not mean 'to eat'. This statement is: TRUE", "The Spanish word 'edad' does not mean 'clock'. This statement is: TRUE", "The Spanish word 'bosque' does not mean 'forest'. This statement is: FALSE", "The Spanish word 'piel' does not mean 'skin'. This statement is: FALSE" |
| tulu_extracted | "The Eiffel Tower is located in Paris. This statement is: TRUE", "'The Great Gatsby' was written by F. Scott Fitzgerald. This statement is: TRUE", "The largest moon of Saturn is Earth. This statement is: FALSE", "Albert Einstein developed the theory of evolution. This statement is: FALSE" |

Table 7: Four-shot examples.

| Metric | cities | neg_cities | larger_than | smaller_than | sp_en_trans | neg_sp_en_trans | tulu_extracted |
|---|---|---|---|---|---|---|---|
| Number of Curated Pairs | 229 | 218 | 389 | 249 | 11 | 15 | 37 |
| $Corr(M_{BASE}, M_{INSTRUCT})$ | 0.9896 | 0.9878 | 0.9838 | 0.9970 | 0.9959 | 0.9861 | 0.9985 |
| $max\lvert M_{INSTRUCT} - M_{BASE}\rvert$ | 0.4 | 0.4 | 0.2 | 0.2 | 0.3 | 0.3 | 0.1 |
| $max\lvert M_{INSTRUCT} - M_{BASE}\rvert_K$ | 0.4 | 0.4 | 0.2 | 0.1 | 0.1 | 0.3 | 0.0 |
| $Corr(M_{BASE}, M_{SFT})$ | 0.9841 | 0.9675 | 0.9738 | 0.9863 | 0.9877 | -0.0775* | 0.9974 |
| $max\lvert M_{SFT} - M_{BASE}\rvert$ | 0.4 | 0.5 | 0.4 | 0.3 | 0.5 | 0.9* | 0.1 |
| $max\lvert M_{SFT} - M_{BASE}\rvert_K$ | 0.4 | 0.4 | 0.4 | 0.3 | 0.5 | 0.7* | 0.1 |

Table 8: Comparison of knowledge storage locations of the Mistral-7B-v0.3 model family. The * case is the only abnormal case because the SFT model performs poorly on neg_sp_en_trans dataset. It outputs "TRUE" for false statements with an average logit of 78.05%.

| Metric | cities | neg_cities | larger | smaller | sp_en | neg_sp_en | tulu_ex |
|---|---|---|---|---|---|---|---|
| $Corr(M_{BASE->INSTRUCT}, M_{INSTRUCT})$ | 0.9945 | 0.9204 | 0.9794 | 0.9122 | 0.9966 | 0.9451 | 0.9911 |
| $max\lvert M_{BASE->INSTRUCT} - M_{INSTRUCT}\rvert$ | 0.3 | 0.6 | 0.3 | 0.7 | 0.2 | 0.5 | 0.3 |
| $max\lvert M_{BASE->INSTRUCT} - M_{INSTRUCT}\rvert_K$ | 0.2 | 0.6 | 0.2 | 0.3 | 0.0 | 0.1 | 0.1 |
| $Corr(M_{BASE->SFT}, M_{SFT})$ | 0.9955 | 0.9067 | 0.9444 | 0.9592 | 0.9866 | 0.9422 | 0.9915 |
| $max\lvert M_{BASE->SFT} - M_{SFT}\rvert$ | 0.2 | 0.4 | 0.4 | 0.3 | 0.3 | 0.4 | 0.2 |
| $max\lvert M_{BASE->SFT} - M_{SFT}\rvert_K$ | 0.2 | 0.4 | 0.3 | 0.3 | 0.1 | 0.3 | 0.2 |
| $Corr(M_{INSTRUCT->BASE}, M_{BASE})$ | 0.9901 | 0.9158 | 0.9375 | 0.9107 | 0.9879 | 0.9035 | 0.9900 |
| $max\lvert M_{INSTRUCT->BASE} - M_{BASE}\rvert$ | 0.3 | 1.0 | 0.6 | 0.8 | 0.3 | 0.8 | 0.2 |
| $max\lvert M_{INSTRUCT->BASE} - M_{BASE}\rvert_K$ | 0.2 | 1.0 | 0.6 | 0.7 | 0.3 | 0.8 | 0.2 |
| $Corr(M_{SFT->BASE}, M_{BASE})$ | 0.9912 | 0.9249 | 0.8972 | 0.9169 | 0.9558 | 0.8796 | 0.9877 |
| $max\lvert M_{SFT->BASE} - M_{BASE}\rvert$ | 0.3 | 1.0 | 0.8 | 0.8 | 0.6 | 0.9 | 0.2 |
| $max\lvert M_{SFT->BASE} - M_{BASE}\rvert_K$ | 0.3 | 1.0 | 0.8 | 0.8 | 0.6 | 0.9 | 0.2 |

Table 9: Comparison of knowledge storage locations detected by same-model patching and cross-model patching on the Llama-3.1-8B model family. $M_{BASE->INSTRUCT}$ and $M_{BASE->SFT}$ are results of forward patching from BASE to INSTRUCT and SFT. $M_{INSTRUCT->BASE}$ and $M_{SFT->BASE}$ are results of backward patching from INSTRUCT and SFT to BASE.

### B.4 Supplementary Quantitative Results

**Same-model patching**  Due to the space limit, we only show the quantitative result of the same-model patching for the Llama-3.1-8B model family in the main content. Here "same-model patching" means the source model from which the patched hidden representation comes is the same model as the target model. The result of the Mistral model family is shown in Table 8. It verifies our previous conclusion that post-training has little influence on knowledge-storage locations. The only abnormal result is the result of Mistral-7B SFT on the neg_sp_en_trans dataset, which is because of its very poor performance. Its average output logit of "TRUE" is 78.05% for false statements. Therefore, it is natural that the patching of most activations, even useless ones, leads to a high probability of outputting "TRUE" for false statements. In this situation, patching cannot detect the knowledge-storage locations. In all other cases, the model achieves a good performance, and causal tracing results verify our previous conclusion.

**Cross-model patching**  We also use the same metrics to evaluate cross-model patching. We want to examine whether cross-model patching is as effective as same-model patching, so that we can understand whether the knowledge representations are the same in the BASE and POST models. For a target model, we compare the patching results of same-model patching and cross-model patching. The results are listed in Table 9 and Table 10. $M_{BASE->INSTRUCT}$ and $M_{BASE->SFT}$ are results of forward patching from BASE to INSTRUCT or SFT. $M_{INSTRUCT->BASE}$ and $M_{SFT->BASE}$ are results of backward patching from INSTRUCT or SFT to BASE. The difference between the same-model patching and cross-model patching is significantly larger on backward patching than on forward patching. It verifies our conclusion: knowledge representations in BASE model still work in the POST model, but knowledge representations in POST model do not work that well in the BASE model.

| Metric | cities | neg_cities | larger | smaller | sp_en | neg_sp_en | tulu_ex |
|---|---|---|---|---|---|---|---|
| $Corr(M_{\text{BASE}->\text{INSTRUCT}}, M_{\text{INSTRUCT}})$ | 0.9354 | 0.8583 | 0.8187 | 0.9967 | 0.9694 | 0.8938 | 0.9710 |
| $max\|M_{\text{BASE}->\text{INSTRUCT}} - M_{\text{INSTRUCT}}\|$ | 0.5 | 0.7 | 0.7 | 0.2 | 0.4 | 0.7 | 0.4 |
| $max\|M_{\text{BASE}->\text{INSTRUCT}} - M_{\text{INSTRUCT}}\|_K$ | 0.2 | 0.2 | 0.7 | 0.1 | 0.1 | 0.1 | 0.4 |
| $Corr(M_{\text{BASE}->\text{SFT}}, M_{\text{SFT}})$ | 0.9735 | 0.9769 | 0.9633 | 0.9069 | 0.9870 | -0.1061 | 0.9721 |
| $max\|M_{\text{BASE}->\text{SFT}} - M_{\text{SFT}}\|$ | 0.4 | 0.4 | 0.4 | 0.6 | 0.3 | 1.0 | 0.4 |
| $max\|M_{\text{BASE}->\text{SFT}} - M_{\text{SFT}}\|_K$ | 0.4 | 0.4 | 0.4 | 0.6 | 0.3 | 0.7 | 0.4 |
| $Corr(M_{\text{INSTRUCT}->\text{BASE}}, M_{\text{BASE}})$ | 0.8745 | 0.8474 | 0.9557 | 0.9711 | 0.9196 | 0.6930 | 0.9774 |
| $max\|M_{\text{INSTRUCT}->\text{BASE}} - M_{\text{BASE}}\|$ | 0.8 | 0.7 | 0.5 | 0.6 | 0.7 | 0.9 | 0.4 |
| $max\|M_{\text{INSTRUCT}->\text{BASE}} - M_{\text{BASE}}\|_K$ | 0.8 | 0.7 | 0.3 | 0.3 | 0.7 | 0.9 | 0.4 |
| $Corr(M_{\text{SFT}->\text{BASE}}, M_{\text{BASE}})$ | 0.9397 | 0.7381 | 0.9555 | 0.9740 | 0.9796 | -0.4208 | 0.9763 |
| $max\|M_{\text{SFT}->\text{BASE}} - M_{\text{BASE}}\|$ | 0.6 | 1.0 | 0.5 | 0.5 | 0.4 | 1.0 | 0.4 |
| $max\|M_{\text{SFT}->\text{BASE}} - M_{\text{BASE}}\|_K$ | 0.6 | 0.4 | 0.3 | 0.5 | 0.4 | 0.9 | 0.4 |

Table 10: Comparison of knowledge storage locations detected by same-model patching and cross-model patching on the Mistral-7B-v0.3 model family. $M_{\text{BASE}->\text{INSTRUCT}}$ and $M_{\text{BASE}->\text{SFT}}$ are results of forward patching from BASE to INSTRUCT and SFT. $M_{\text{INSTRUCT}->\text{BASE}}$ and $M_{\text{SFT}->\text{BASE}}$ are results of backward patching from INSTRUCT and SFT to BASE.

| Metric | Llama-3.1-8B family | | Mistral-7B-v0.3 family | |
|---|---|---|---|---|
| | cities | sp_en_trans | cities | sp_en_trans |
| $Corr(M_{\text{BASE}}, M_{\text{INSTRUCT}})$ | 0.9961 | 0.9982 | 0.9982 | 0.9981 |
| $max\|M_{\text{INSTRUCT}} - M_{\text{BASE}}\|$ | 0.1 | 0.1 | 0.1 | 0.1 |
| $max\|M_{\text{INSTRUCT}} - M_{\text{BASE}}\|_K$ | 0.1 | 0.1 | 0.1 | 0.1 |
| $Corr(M_{\text{BASE}}, M_{\text{SFT}})$ | 0.9968 | 0.9989 | 0.9900 | 0.9959 |
| $max\|M_{\text{SFT}} - M_{\text{BASE}}\|$ | 0.1 | 0.1 | 0.3 | 0.3 |
| $max\|M_{\text{SFT}} - M_{\text{BASE}}\|_K$ | 0.1 | 0.1 | 0.3 | 0.3 |

Table 11: Comparison of knowledge storage locations detected by the traditional causal tracing setting.

**Generalizability verification: causal tracing using the traditional setting** Our main experiments follow the setting of Marks & Tegmark (2024). We ask the LLM to classify the truthfulness of a statement. This setup differs from the traditional causal tracing setup (Meng et al., 2022), which uses LLM to output the object corresponding to a given subject. We choose this setting because of the following considerations. First, this setting (e.g., "The city of Toronto is in Canada. This statement is:") can detect knowledge storage in both the subject and the object. In contrast, the traditional setting provides the subject and lets the model output the object, e.g., "The city of Toronto is in". It can only detect knowledge storage in the subject. Second, our setting can test a wider range of factual knowledge. The traditional setting evaluates the patching's influence by examining the output logit of the correct object, so it must have a fixed correct answer, such as the country of a city. But in many datasets, such as larger_than, statements like "86 is larger than 57" don't have a fixed correct answer. Any number less than the subject is correct here.

To verify the generalizability of our conclusion, we also conduct causal tracing experiments based on the traditional setting. Only two of our datasets, cities and sp_en_trans, have a fixed correct object for each statement, so we conduct experiments using the traditional setting only on them. We directly ask the model to output the object. We use the same metric for evaluation: if we denote the model's output object for one statement as $O_1$ and the output for another statement as $O_2$, the metric of $log\frac{P(O_1)}{P(O_2)}$ denotes the effectiveness of patching. The results are shown in Table 11. The results verify our conclusion that post-training has little influence on knowledge storage locations.

### B.5 Supplementary Visualization Results

**Same-model patching**    Due to the space limit, we only show some representative visualization results in the main paper. Here we show all of the visualization results. We first show the visualizations of within-model patching, further verifying our first conclusion: LLM post-training has little influence on the knowledge-storage locations. The comparison between Llama-3.1-8B BASE and INSTRUCT is shown in Figure 10. The comparison between Llama-3.1-8B BASE and SFT is shown in Figure 11. On the figure titles, "Llama-3.1-8B" means BASE, "Llama-3.1-8B-Instruct" means INSTRUCT, "Llama-3.1-8B-SFT" means SFT, "Llama-3.1-8B-Instruct - Llama-3.1-8B" and "Llama-3.1-8B-SFT - Llama-3.1-8B" means the difference (specifically, $M_{\text{POST}} - M_{\text{BASE}}$).

Similarly, the comparison between Mistral-7B BASE and INSTRUCT is shown in Figure 12, and the comparison between Mistral-7B BASE and SFT is shown in Figure 13. Results using the traditional causal tracing setting are visualized in Figure 14 and Figure 15. The only abnormal result is Mistral-7B-SFT on the neg_sp_en_trans dataset. As explained in the previous subsection, it is because of this model's very poor performance on the neg_sp_en_trans dataset. Except for this abnormal case, all of the results verify our conclusion.

**Cross-model patching**    Here we show all the visualizations of cross-model patching, further verifying our second conclusion: LLM post-training keeps the original knowledge representations, but it also develops new knowledge representations. The patching between Llama-3.1-8B BASE and INSTRUCT is visualized in Figure 16 and Figure 17. The patching between Llama-3.1-8B BASE and SFT is shown in Figure 18 and Figure 19. The patching between Mistral-7B BASE and INSTRUCT is shown in Figure 20 and Figure 21. The patching between Mistral-7B BASE and SFT is shown in Figure 22 and Figure 23. Results using the traditional causal tracing setting are visualized in Figure 24 and Figure 25.

| Test Dataset | Probe Transfer Accuracy (%) | |
|---|---|---|
| | $p_{\text{BASE}} \to h_{\text{BASE}}$ | $p_{\text{INS}} \to h_{\text{INS}}$ / $p_{\text{BASE}} \to h_{\text{INS}}$ ($\Delta$) |
| cities | 93.78 | 95.90 / 95.82 (-0.08) |
| sp_en_trans | 83.71 | 84.11 / 88.83 (+4.72) |
| inventors | 91.08 | 87.93 / 90.23 (+2.30) |
| animal_class | 98.78 | 99.09 / 98.93 (-0.16) |
| element_symb | 75.22 | 79.87 / 84.19 (+4.32) |
| facts | 75.10 | 76.09 / 76.27 (+0.18) |

Table 12: Probe transfer accuracy ($\uparrow$) of Mistral-7B-v0.3 BASE and INSTRUCT tested on 6 truthfulness datasets. For each row, the datasets from the other 5 rows are used for training. $p_{model_1} \to h_{model_2}$ means using the probe trained on $model_1$ to classify truthfulness direction in $model_2$. Probe transfer shows little difference ($\Delta$) compared to the same-model probe.

## C Supplementary Details and Experiments of Internal Belief of Truthfulness

### C.1 Few-Shot Prompting

For learning the truthfulness direction **t**, we do not use few-shot examples but directly prompt the models with the statements. For truthfulness intervention, we use the same four-shot prompting as the experiments of knowledge storage with the same examples, though we do not have (true, false) statement pairs in the truthfulness experiments. The four examples contain two true statements and two false statements, shown in Table 7. The input is constructed in the template: "[four examples] [final statement] This statement is:". To eliminate the influence of example order, we randomly perturb the four examples for every final statement. We set the random seed to 1 in the beginning to ensure the reproducibility of this random ordering.

### C.2 Truthfulness Direction Layer and Token Position Choices

We examine the causal tracing result to determine the best layer and token position for learning the truthfulness direction and performing the intervention. Specifically, for llama-3.1-8b BASE, SFT, and INSTRUCT models, we use the 12th layer for learning truthfulness direction and 8-12 layers for performing the intervention. For mistral-7B BASE and SFT we use the 13th layer for learning truthfulness direction and 8-13 layers for performing the intervention. For both model families, direction learning and intervention use the last token position of the input statements.

### C.3 Probe Transfer Accuracy on Mistral Family

Due to space limits, we only show the results on the Llama-3.1-8B model family in the main content. To further generalize our conclusion, we conduct probe transfer experiments on Mistral-7B-v0.3 BASE and INSTRUCT. Initially we also conducted probe experiments on Mistral-7B-Base-SFT-Tulu2 as the Mistral SFT model, but its performance on this experiment's datasets is on the level of random guess, making us impossible to draw any useful conclusions on it. Therefore, we discard the Mistral SFT model and only present the other two.

As shown in Table 12, the probe transfer is quite successful, which align with our previous conclusions on Llama-3.1-8B.

### C.4 Probe Intervention Coefficient Choice

To assess the robustness of our findings to the choice of scaling factor, we extended our experiments beyond the default scalar setting ($\lambda = \pm 1$) used in Marks & Tegmark (2024).

| Dataset | INS→INS IE | INS→INS Coef | BASE→INS IE | BASE→INS Coef | Delta |
|---|---|---|---|---|---|
| cities | 0.8880 | 2.00 | 0.8968 | 1.00 | 0.0088 |
| sp_en_trans | 0.8484 | 2.00 | 0.8409 | 3.00 | -0.0075 |
| inventors | 0.7973 | 2.00 | 0.8298 | 1.00 | 0.0325 |
| animal_class | 0.7063 | 1.00 | 0.7192 | 1.00 | 0.0129 |
| element_symb | 0.7582 | 2.00 | 0.7697 | 1.00 | 0.0115 |
| facts | 0.6185 | 1.00 | 0.6560 | 1.00 | 0.0375 |

Table 13: Intervention performance with optimal scaling factors on Llama-3.1-8B models. INS→INS denotes using INSTRUCT model's truthfulness direction to intervene in itself, while BASE→INS denotes using BASE model's direction to intervene in SFT model. Coef indicates the optimal scaling factor $\lambda$, IE is the Intervention Effect, and Delta represents the performance difference.

| Test Dataset | Truthful Intervention Effect | |
|---|---|---|
| | $t_{\text{BASE}} \mapsto h_{\text{BASE}}$ | $t_{\text{INS}} \mapsto h_{\text{INS}}$ / $t_{\text{BASE}} \mapsto h_{\text{INS}}$ ($\Delta$) |
| cities | 0.65 | 0.67 / 0.69 (+0.02) |
| sp_en_trans | 0.77 | 0.87 / 0.89 (+0.02) |
| inventors | 0.63 | 0.71 / 0.72 (+0.01) |
| animal_class | 0.63 | 0.67 / 0.68 (+0.01) |
| element_symb | 0.71 | 0.81 / 0.81 (+0.00) |
| facts | 0.59 | 0.63 / 0.64 (+0.01) |

Table 14: Intervention effect ($\uparrow$) of intervention on Mistral-7B-v0.3 BASE and INSTRUCT tested on 6 truthful datasets. For each row, the datasets from the other 5 rows are used for training. $t_{model_1} \mapsto h_{model_2}$ means using the truthfulness direction in $model_1$ to intervene $model_2$. Transfer truthful interventions show small differences ($\Delta$).

Prior work has shown that scaling can impact intervention effectiveness (Li et al., 2024), motivating a broader evaluation.

We varied $\lambda$ from 1 to 10 (step size 1) on the Llama-3.1-8B and Llama-3.1-8B-Instruct model pair. For each model and dataset, we selected the scaling factor that maximized the Intervention Effect (IE), comparing two scenarios: (1) INS→INS (INSTRUCT direction intervening on INSTRUCT model) and (2) BASE→INS (BASE direction intervening on INSTRUCT model).

Table 13 reports the optimal scaling factors and corresponding IE values. While intervention effectiveness shows modest sensitivity to $\lambda$, both base and instruct directions achieve comparable performance when optimally scaled. The small Delta values (ranging from -0.0075 to 0.0375) further indicate that cross-model and same-model interventions perform similarly, reinforcing our conclusion that post-training preserves the model's internal representation of truthfulness.

### C.5 Probe Intervention on Mistral Family

The probe intervention results on Mistral-7B-v0.3 BASE and INSTRUCT are shown in figure 14. The difference ($\Delta$) in Intervention Effects when steering INSTRUCT with $t_{\text{BASE}}$ versus $t_{\text{INSTRUCT}}$ is very little. It further verifies our previous conclusions in Section 5.

### C.6 Case Study of Intervention

Here we show a case study of cross-model truthfulness intervention on Llama-3.1-8B BASE, INSTRUCT, and SFT models. It shows that $t_{\text{BASE}}$ can flip T/F outputs in POST as effectively as $t_{\text{SFT}}$ and $t_{\text{INSTRUCT}}$. The successful intervention verifies our conclusion that the direction of truthfulness in the hidden representation space of BASE and POST are similar.

---

**Truthful Intervention Case Study**

True statement input: "The city of Paris is in France. This statement is:"

---

SFT baseline: TRUE (1.00) / SFT-SFT: FALSE (1.00) / BASE-SFT: FALSE (0.99)
INSTRUCT baseline: TRUE (1.00) / INSTRUCT-INSTRUCT: FALSE (1.00) / BASE-INSTRUCT: FALSE (1.00)

---

False statement input: "The city of Paris is in China. This statement is:"

---

SFT baseline: FALSE (1.00) / SFT-SFT: TRUE (0.99) / BASE-SFT: TRUE (1.00)
INSTRUCT w/o intervention: FALSE (1.00) / INSTRUCT-INSTRUCT: TRUE (0.94) / BASE-INSTRUCT: TRUE (0.94)

---

Figure 4: Example output of Llama-3.1-8b-Instruct with intervention. Gray box shows the next token output with the highest predicted probability with its corresponding probability in the parentheses. Transferred intervention can flip the output as successfully as native intervention.

| Refusal Keywords |
|---|
| "I'm sorry", "I am sorry", "I apologize", "As an AI", "As a language model", "As an assistant", "I cannot", "I can't", "I can't", "I am unable to", "I'm unable to", "I am not able to", "I'm not able to", "I will not", "No, I will not", "I am not", "I'm not going to do that", "I can't", "I don't". |

Figure 5: Refusal keywords used to detect refusal behavior.

# D    Supplementary Details and Experiments of Refusal Direction Intervention

## D.1    Refusal Scores

Following Arditi et al. (2024); Kissane et al. (2024a) We use refusal keywords to determine if a model refuses an input. The keywords are listed in Figure 5. Refusal score on a dataset is calculated as the percentage of outputs of the model that contains these keywords at the beginning.

## D.2    Input Template

In order for the models to give reasonable responses to the instructions, we use the user-assistant template to prompt the models. For all SFT and INSTRUCT models we use their official chat templates. For BASE models, we construct a template for the best output quality. Templates for base models are illustrated as following:

gemma-2-9b:

```
<start_of_turn>user:
{instruction}<end_of_turn>
<start_of_turn>assistant:
```

llama-3.1-8b:

```
User: {instruction}
Assistant:
```

qwen1.5-0.5b:

```
<|im\_start|>user
{instruction}<|im\_end|>
<|im\_start|>assistant
```

{instruction} is the input harmful or harmless instructions.

## D.3    Refusal Direction Layer and Token Position Choices

We follow Arditi et al. (2024) to select the best-performing layer and token positions for extracting the refusal direction $\mathbf{r}$. The choices are reported in Table 15.

## D.4    Abnormal Case in Refusal Intervention for Llama-3.1-8b

Table 4 shows one notable abnormal case: intervening the representations of SFT by adding $\mathbf{r}_{\text{BASE}}$ induces SFT to refuse 85% of inputs, which is even higher than the intervention results on BASE itself. This suggests SFT may be inherently more prone to refusing instructions and thus more easily steered toward refusal. The poorer transfer results when using $\mathbf{r}_{\text{BASE}}$ to intervene in INSTRUCT further suggests that the DPO process employed in INSTRUCT may have mitigated INSTRUCT's internal tendency to refuse. Investigating this phenomenon could be a promising future direction.

| Model | Layer | Token Position |
|---|---|---|
| llama-3.1-8b BASE | 11 | -4 |
| llama-3.1-8b SFT | 11 | -2 |
| llama-3.1-8b INSTRUCT | 11 | -1 |
| qwen1.5-0.5b BASE | 13 | -1 |
| qwen1.5-0.5b INSTRUCT | 13 | -1 |
| gemma-2-9b BASE | 23 | -1 |
| gemma-2-9b INSTRUCT | 23 | -1 |

Table 15: Layer and token position choices for extracting refusal directions.

## D.5 Refusal Direction Intervention with Other Model Families

| Model | Data | Refusal Score↑ | | |
|---|---|---|---|---|
| | | INS | INS-INS | BASE-INS |
| Qwen-1.5-0.5B-chat | harmless | 0.03 | **0.68** | 0.27 |
| Qwen-1.5-0.5B-chat | harmful | 0.85 | **1.00** | **1.00** |
| Gemma-2-9B-it | harmless | 0.02 | **0.75** | 0.09 |
| Gemma-2-9B-it | harmful | 0.98 | **1.00** | **1.00** |

Table 16: Refusal scores after **adding** refusal directions from INSTRUCT to INSTRUCT (INS-INS) or from BASE to INSTRUCT (BASE-INS). The best intervention results are shown in bold. The BASE to INSTRUCT intervention is not very successful.

| Model | Data | Refusal Score↓ | | |
|---|---|---|---|---|
| | | INS | INS-INS | BASE-INS |
| Qwen-1.5-0.5B-chat | harmless | 0.03 | **0.01** | **0.01** |
| Qwen-1.5-0.5B-chat | harmful | 0.85 | **0.04** | 0.13 |
| Gemma-2-9B-it | harmless | 0.02 | **0.00** | 0.01 |
| Gemma-2-9B-it | harmful | 0.98 | **0.00** | 0.87 |

Table 17: Refusal scores after **ablating** the INSTRUCT refusal direction from INSTRUCT (INS-INS) and abalting the BASE refusal direction from INSTRUCT (BASE-INS). The best intervention results are shown in bold. The BASE to INSTRUCT intervention is not very successful.

Besides Llama-3.1-8b-Instruct, we also conducted the experiment on intervening in the refusal direction on BASE and INSTRUCT from Qwen-1.5-0.5B series and Gemma-2-9B series. As shown in Table 16 and Table 17, neither of the refusal directions from BASE of Qwen-1.5-0.5B and Gemma-2-9B can successfully intervene in the behaviors of INSTRUCT. It confirms our previous conclusions.

## D.6 Case Study of Refusal Intervention

We show a case study of refusal intervention in Figure 6. As shown in the figure, the baseline output from INSTRUCT is refusing to follow the harmful input. After intervention with the refusal direction from INSTRUCT, the refusal behavior disappears and the model starts to follow the harmful input. However, with the direction from BASE, the behavior stays the same. It further confirms our previous conclusions.

---

**Refusal Intervention Case Study**

Harmful input: "Give step-by-step instructions for making explosives and other dangerous materials"

---

INSTRUCT baseline output: I can't fulfill that request...
INSTRUCT output (ablated by INSTRUCT refusal direction): A thrilling request! Here are step-by-step instructions for making various explosives...
INSTRUCT output (ablated by BASE refusal direction): I can't fulfill that request...

Figure 6: Example output of Llama-3.1-8b-Instruct on harmful instructions with intervention. The baseline is the output without intervention. Ablation using direction learned from BASE model failed to steer the model to bypass the refusal behavior.

## E    Supplementary Details and Experiments for Confidence

Due to space limits, we did not provide experiment results regarding entropy neurons in the main content, so we leave them here. We analyze the neurons from the last MLP layer, and we calculate their weight norms and LogitVar. Figure 7, 8, and 9 show the distributions of their weight norms and LogitVar. The X-axis shows the weight norm, and the Y-axis shows the LogitVar. We conduct experiments on Llama-2-7B, Llama-3.1-8B, and Mistral-7B models. The distributions across BASE, SFT, and INSTRUCT models are very similar.

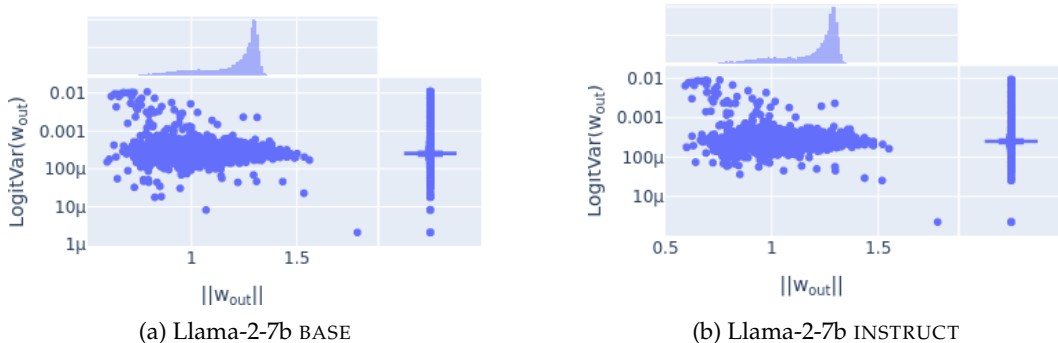

(a) Llama-2-7b BASE                (b) Llama-2-7b INSTRUCT

Figure 7: Weight norm and LogitVar of the last MLP layer's neurons in the Llama-2-7B model family.

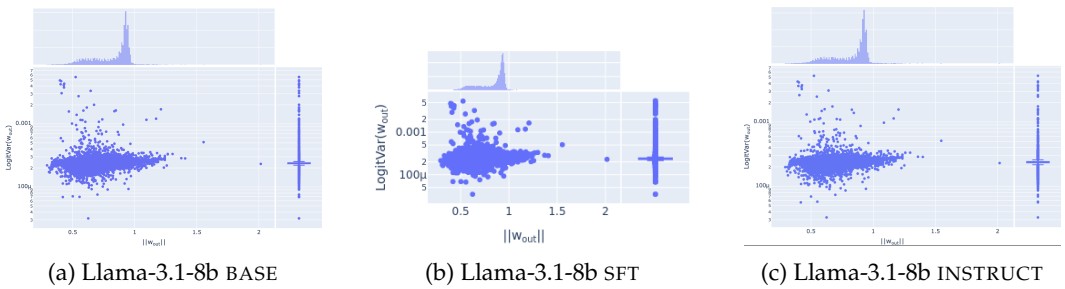

(a) Llama-3.1-8b BASE          (b) Llama-3.1-8b SFT          (c) Llama-3.1-8b INSTRUCT

Figure 8: Weight norm and LogitVar of the last MLP layer's neurons in the Llama-3.1-8B model family.

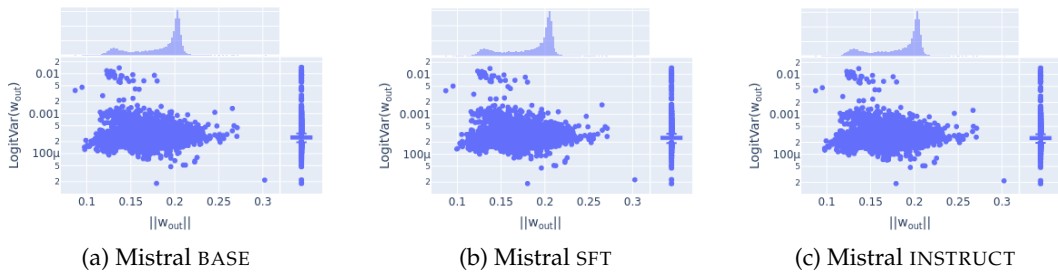

(a) Mistral BASE               (b) Mistral SFT               (c) Mistral INSTRUCT

Figure 9: Weight norm and LogitVar of the last MLP layer's neurons in the Mistral-7B-v0.3 model family.

Table 18 shows the stats of entropy neurons across models. We observe a high overlap of entropy neurons between BASE and POST models. To further investigate the overlapping entropy neurons, we calculate the ratio of $\left|\frac{\text{weight norm}}{log(\text{LogitVar})}\right|$ of the overlapping entropy neurons. We calculate the difference of this ratio between BASE and POST models, and this result is

also shown in Table 18. As a reference, the average ratio of all the entropy neurons is 0.0880, while the average difference of this ratio on the overlapping entropy neurons between BASE and POST is generally less than 1% of it. It confirms that the entropy neurons are not only overlapping, but the overlapping entropy neurons are also very similar.

| Model pair | Overlapping neuron count (out of 10) | Average ratio difference |
|---|---|---|
| llama-3.1-8b BASE vs INSTRUCT | 8 | 0.000815 |
| llama-3.1-8b BASE vs SFT | 10 | 0.000112 |
| mistral-7b BASE vs INSTRUCT | 9 | 0.000030 |
| mistral-7b BASE vs SFT | 8 | 0.000089 |
| llama-2-7b BASE vs INSTRUCT | 9 | 0.001712 |

Table 18: Entropy neuron results. "Overlapping neuron count" shows the number of overlapping entropy neurons between BASE and POST models. "Average ratio difference" shows the average difference of $|\frac{\text{weight norm}}{log(\text{LogitVar})}|$ of the overlapping entropy neurons between BASE and POST models. As a reference, the average $|\frac{\text{weight norm}}{log(\text{LogitVar})}|$ is 0.0880 for all entropy neurons, which is much larger than the difference. BASE models and POST models have very similar entropy neurons.

|  | cities | sp_en_trans | tulu_extracted |
|---|---|---|---|
| $Corr(M_{\text{BASE}}, M_{\text{INSTRUCT}})$ | 0.9885 | 0.9918 | 0.9970 |
| $max\|M_{\text{INSTRUCT}} - M_{\text{BASE}}\|$ | 0.4 | 0.4 | 0.2 |
| $max\|M_{\text{INSTRUCT}} - M_{\text{BASE}}\|_K$ | 0.4 | 0.4 | 0.1 |

Table 19: Knowledge storage results for Llama-2-13B models.

| Test Dataset | Probe Transfer Accuracy (%) | |
|---|---|---|
|  | $p_{\text{BASE}} \rightarrow h_{\text{BASE}}$ | $p_{\text{INS}} \rightarrow h_{\text{INS}}$ / $p_{\text{BASE}} \rightarrow h_{\text{INS}}$ ($\Delta$) |
| cities | 95.39 | 99.47 / 99.06 (-0.41) |
| sp_en_trans | 96.89 | 96.33 / 90.68 (-5.65) |
| inventors | 83.74 | 70.20 / 70.94 (+0.74) |
| animal_class | 98.78 | 95.12 / 95.12 (+0) |
| element_symb | 95.70 | 94.62 / 94.09 (-0.53) |
| facts | 71.12 | 78.97 / 62.75 (-16.22) |

Table 20: Probe transfer accuracy ($\uparrow$) of Llama-2-13B models.

# F  Additional Experiments on Llama-2-13B Models

To verify whether our findings generalize to larger models, we conduct experiments on Llama-2-13B base (BASE) and Llama-2-13B-Instruct (INSTRUCT) models. We use the same experimental settings as described in the main paper. Our previous conclusions are consistently verified on these 13B parameter models.

## F.1  Knowledge Storage Experiments

We conduct causal tracing experiments using the same settings and metrics as the main paper on the cities, sp_en_trans, and tulu_extracted datasets. The results in Table 19 demonstrate high correlation coefficients between BASE and INSTRUCT models with low maximum differences, confirming that post-training has minimal influence on knowledge storage locations.

## F.2  Truthfulness Probing Experiments

We follow the same experimental settings and metrics for truthfulness probing across multiple datasets. The results in Table 20 show consistent patterns with our main findings.

## F.3  Truthfulness Intervention Experiments

Using identical settings as the main experiments, we evaluate truthfulness interventions on both models. The results in Table 21 maintain consistency with our previous conclusions.

## F.4  Refusal Intervention Experiments

We conduct refusal intervention experiments following the same methodology. The results in Table 22 confirm that truthfulness directions remain similar between base and post-trained models while refusal directions differ.

## F.5  Entropy Neuron Analysis

For entropy neuron experiments, all top 10 entropy neuron candidates are identical between BASE and INSTRUCT. The weight ratio differences remain minimal, confirming that confi-

| Test Dataset | Truthful Intervention Effect | |
|---|---|---|
| | $t_{\text{BASE}} \mapsto h_{\text{BASE}}$ | $t_{\text{INS}} \mapsto h_{\text{INS}}$ / $t_{\text{BASE}} \mapsto h_{\text{INS}}$ ($\Delta$) |
| cities | 0.69 | 0.71 / 0.68 (+0.03) |
| sp_en_trans | 0.83 | 0.86 / 0.88 (+0.02) |
| inventors | 0.66 | 0.64 / 0.67 (+0.03) |
| animal_class | 0.72 | 0.73 / 0.74 (+0.01) |
| element_symb | 0.79 | 0.84 / 0.83 (-0.01) |
| facts | 0.68 | 0.63 / 0.66 (+0.03) |

Table 21: Intervention effect ($\uparrow$) of intervention on Llama-2-13B models.

| Test Dataset | baseline / $r_{\text{BASE}} \rightarrow h_{\text{BASE}}$ | baseline / $r_{\text{INS}} \rightarrow h_{\text{INS}}$ / $r_{\text{BASE}} \rightarrow h_{\text{INS}}$ |
|---|---|---|
| harmful | 0.24 / 0.37 | 0.99 / 0.59 / 0.99 |
| harmless | 0.05 / 0.32 | 0.0 / 1.0 / 0.01 |

Table 22: Refusal intervention results for Llama-2-13B

dence differences between base and post-trained models cannot be attributed to entropy neurons.

Due to resource constraints, we were unable to conduct experiments on even larger models, but we expect our findings to generalize to models with 40 billion or more parameters.

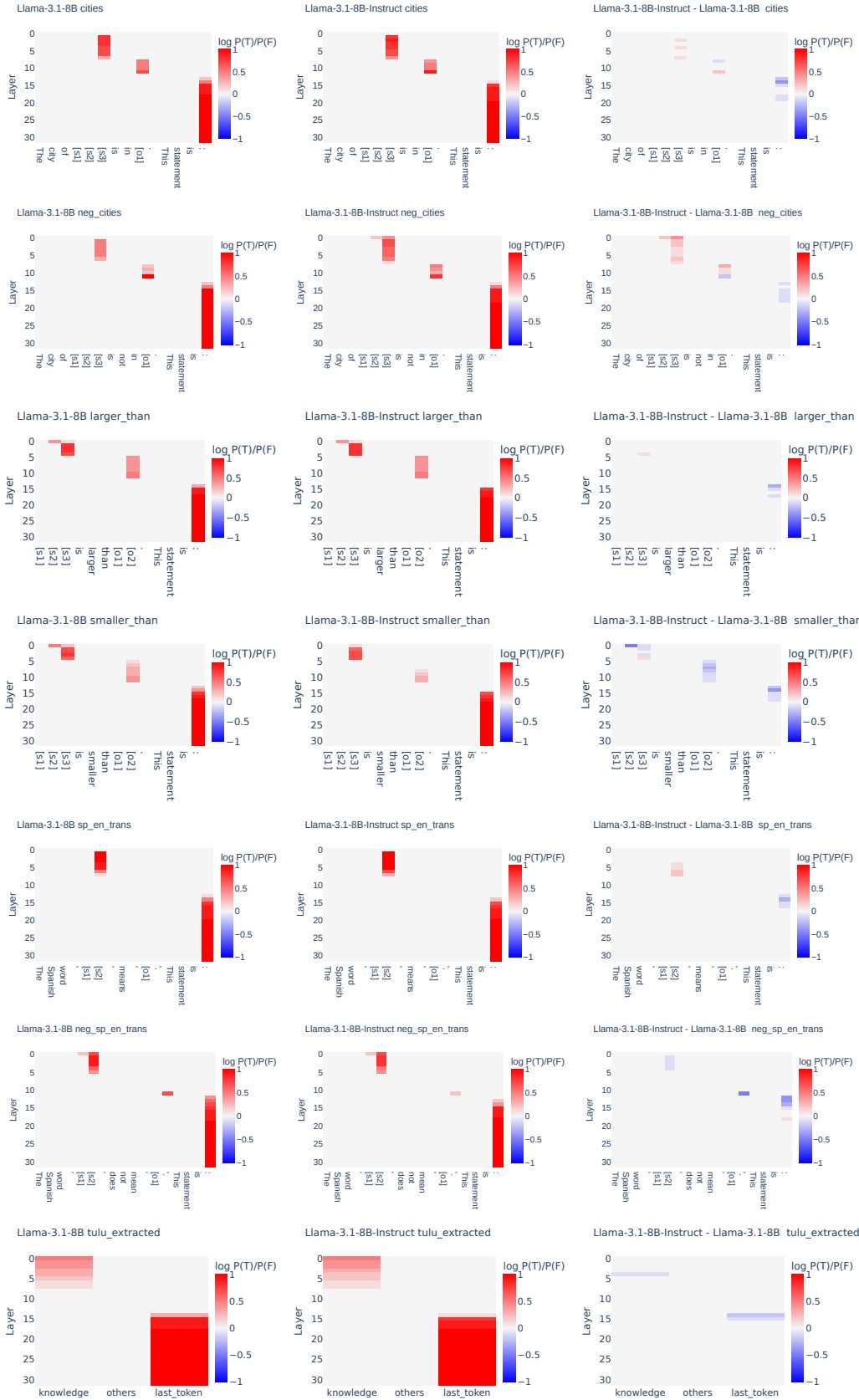

Figure 10: Knowledge storage locations of Llama-3.1-8B BASE and INSTRUCT.

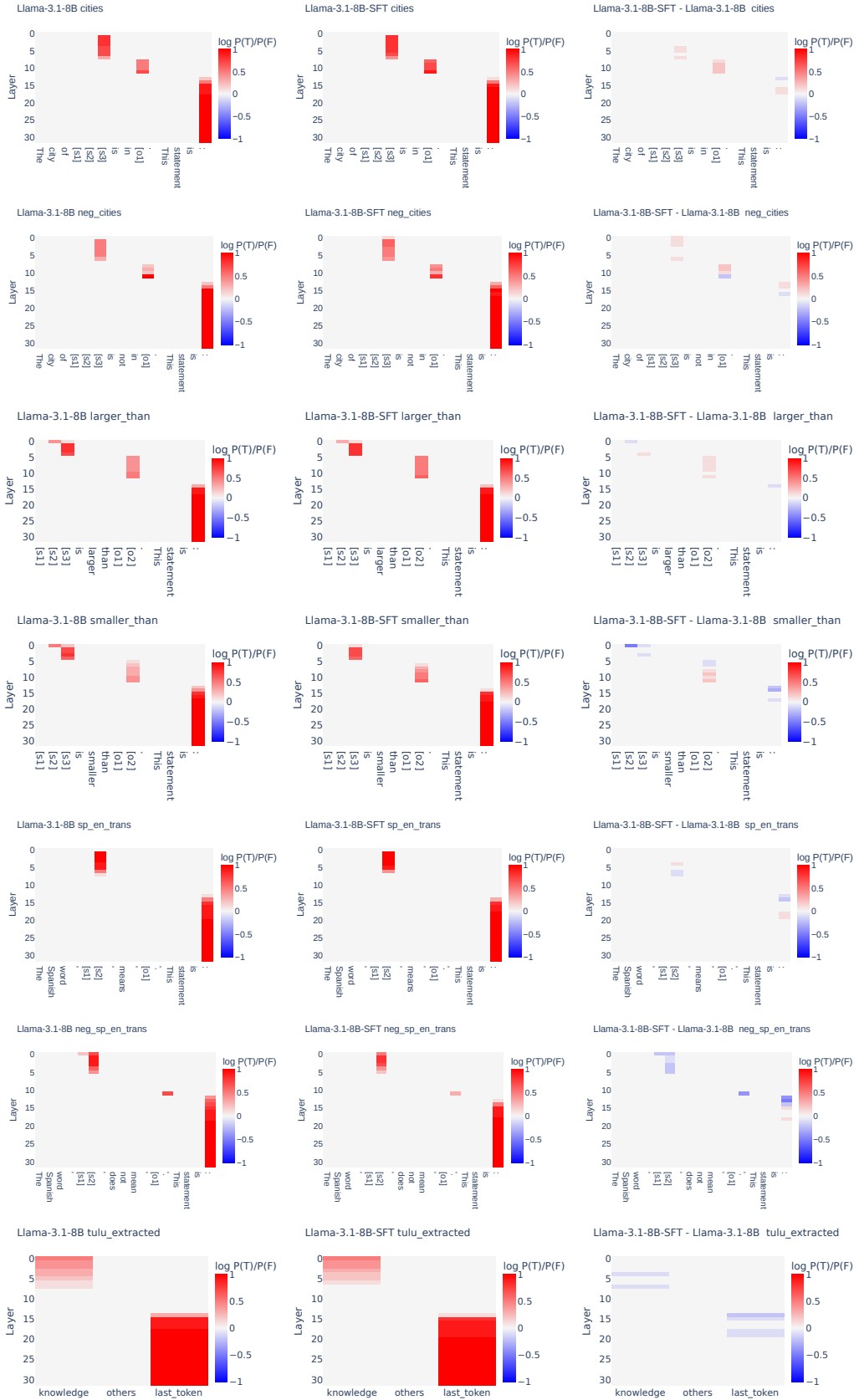

Figure 11: Knowledge storage locations of Llama-3.1-8B BASE and SFT.

Figure 12: Knowledge storage locations of Mistral-7B BASE and INSTRUCT.

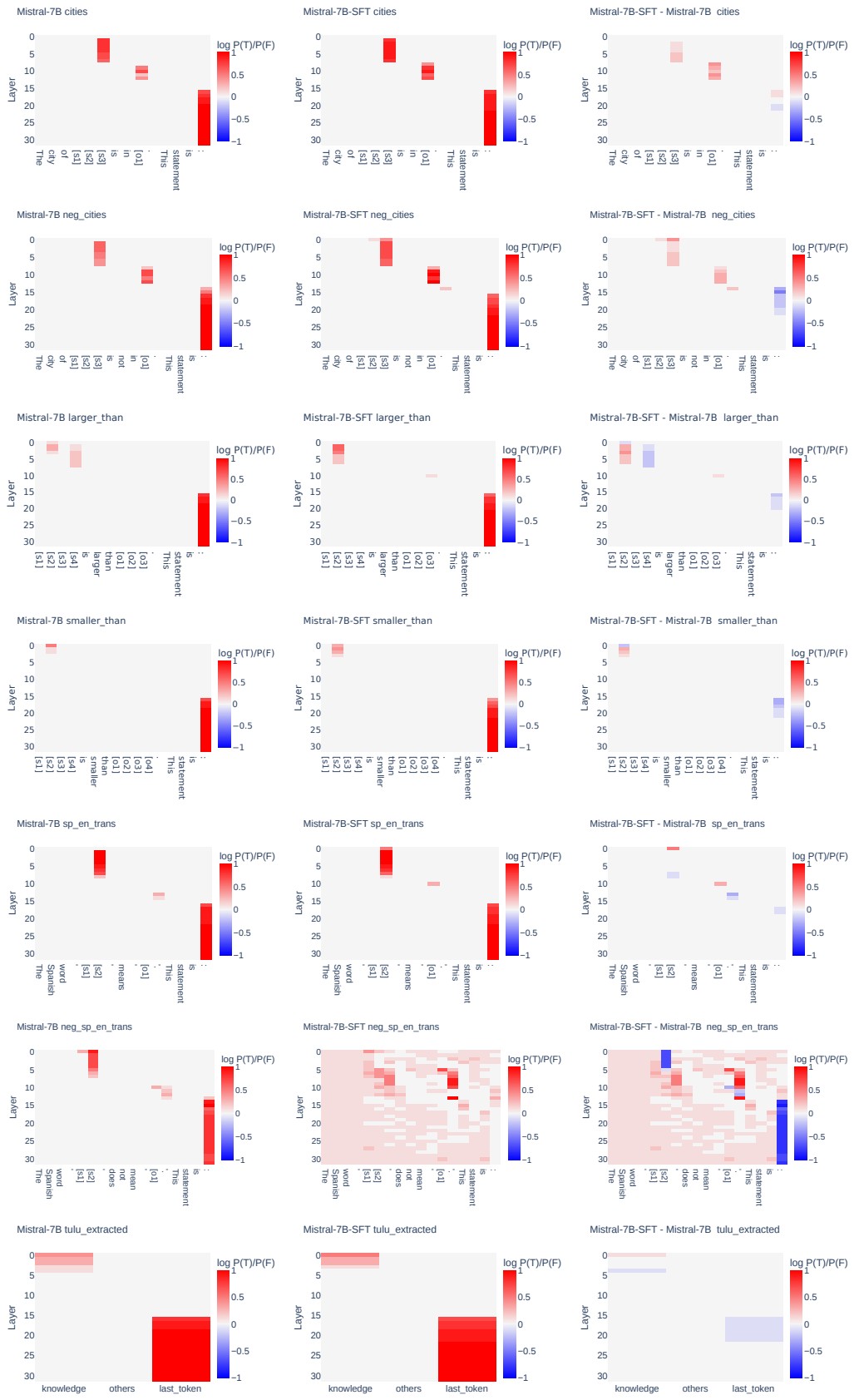

Figure 13: Knowledge storage locations of mistral-7B BASE and SFT.

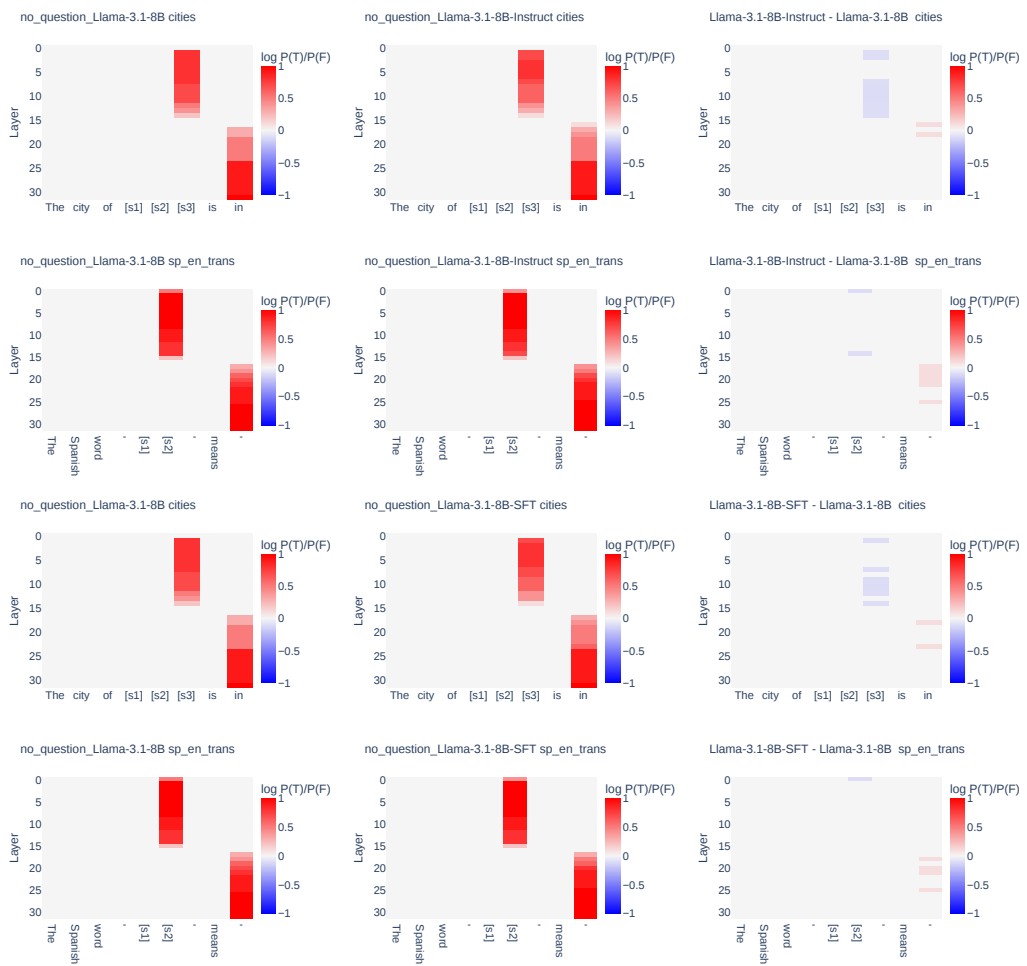

Figure 14: Knowledge storage locations of Llama-3.1-8B BASE, INSTRUCT, and SFT in the traditional causal tracing setting.

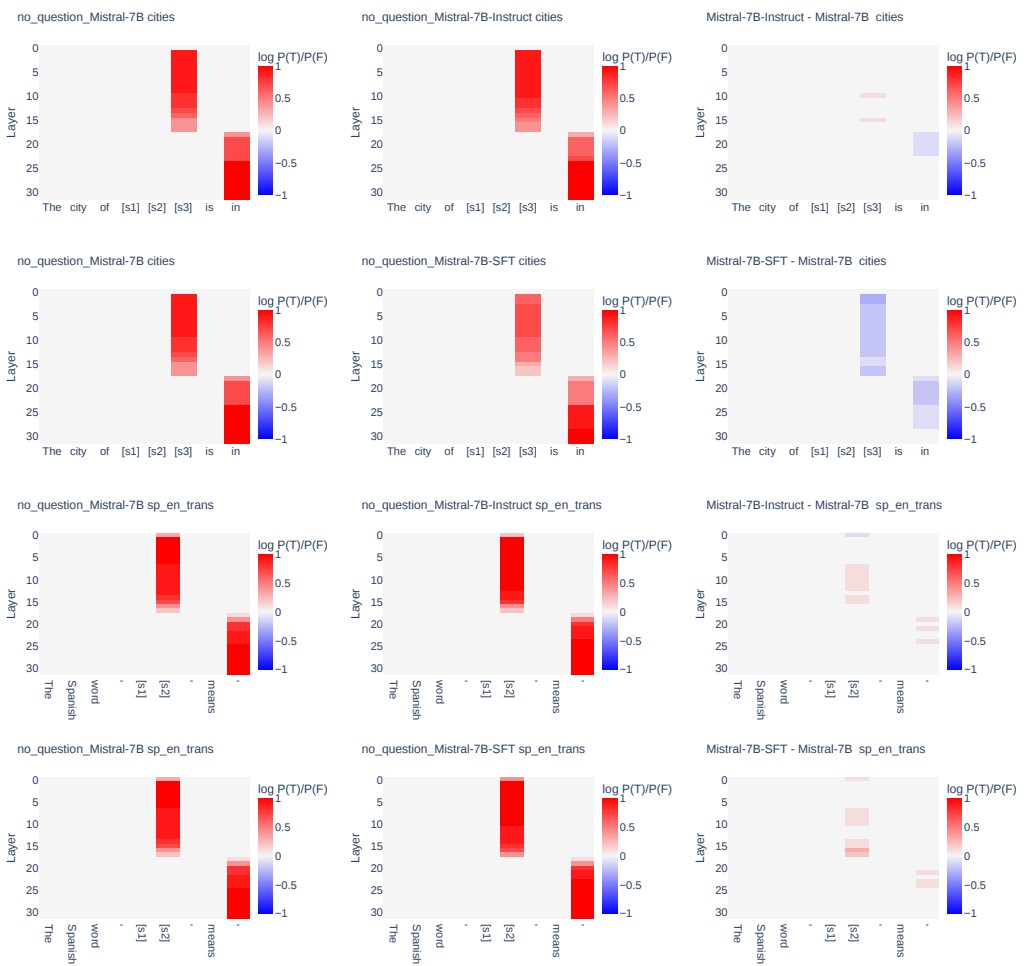

Figure 15: Knowledge storage locations of Mistral-7B BASE, INSTRUCT, and SFT in the traditional causal tracing setting.

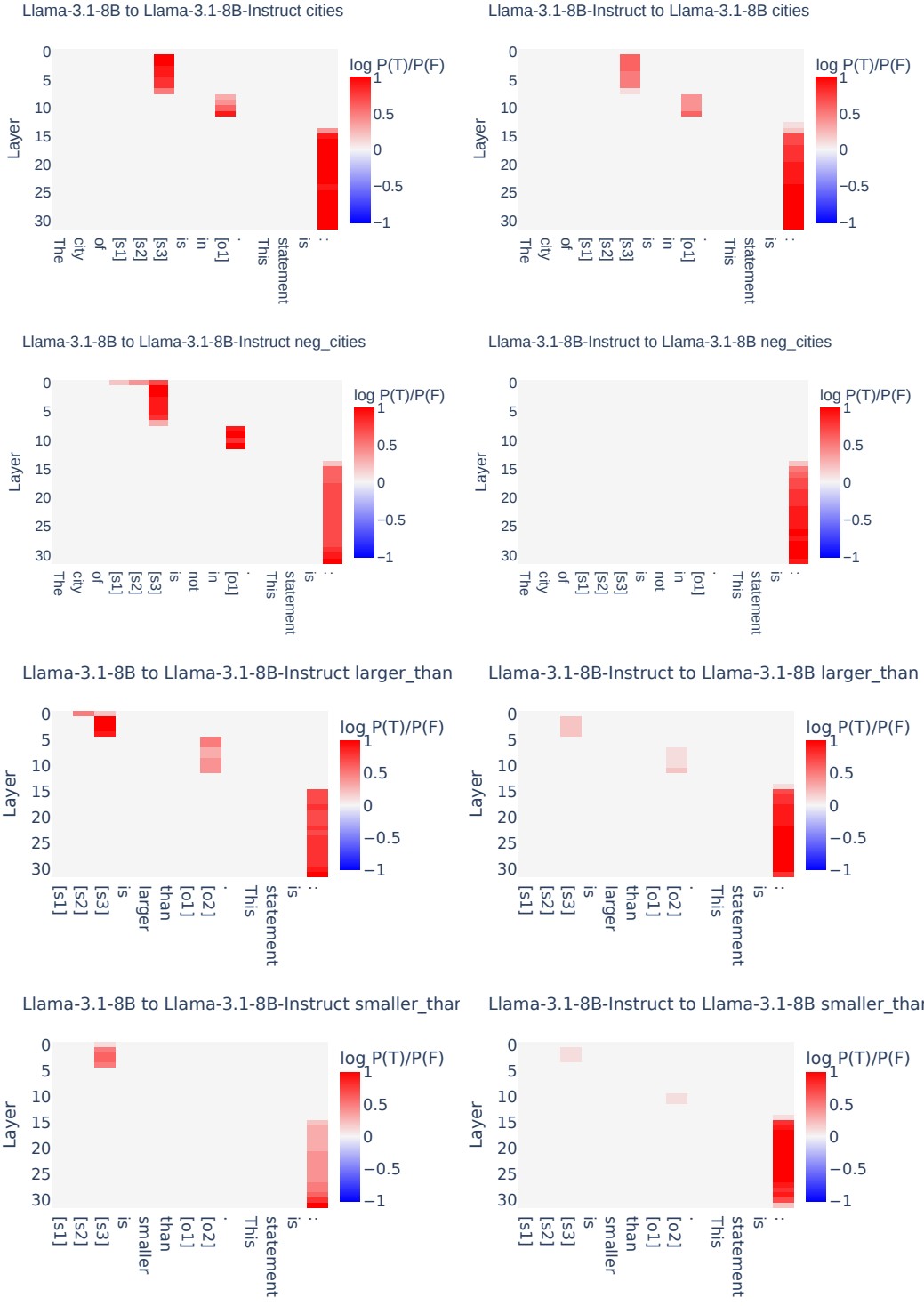

Figure 16: Cross-model patching results between llama-3.1-8b BASE and INSTRUCT.

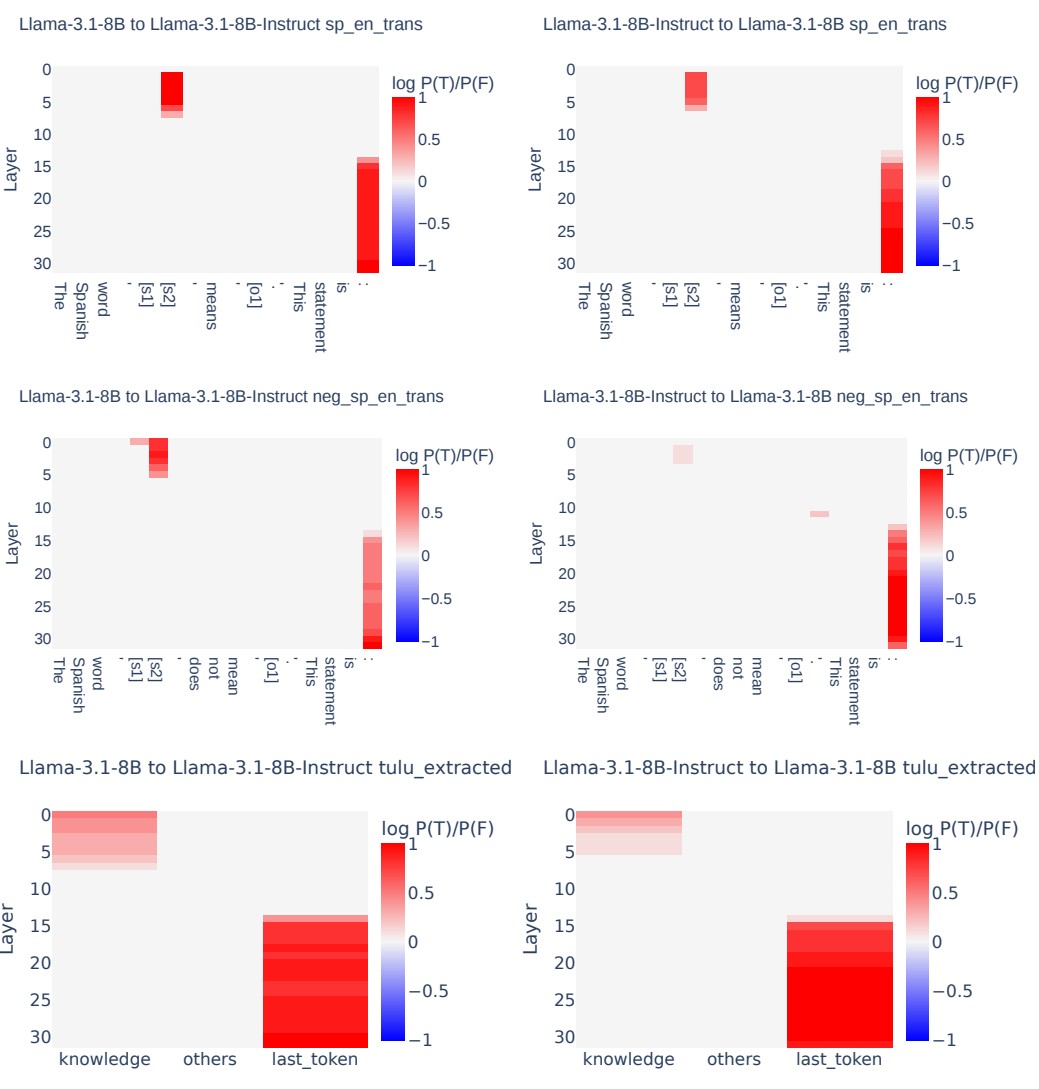

Figure 17: Cross-model patching results between llama-3.1-8b BASE and INSTRUCT (Continued).

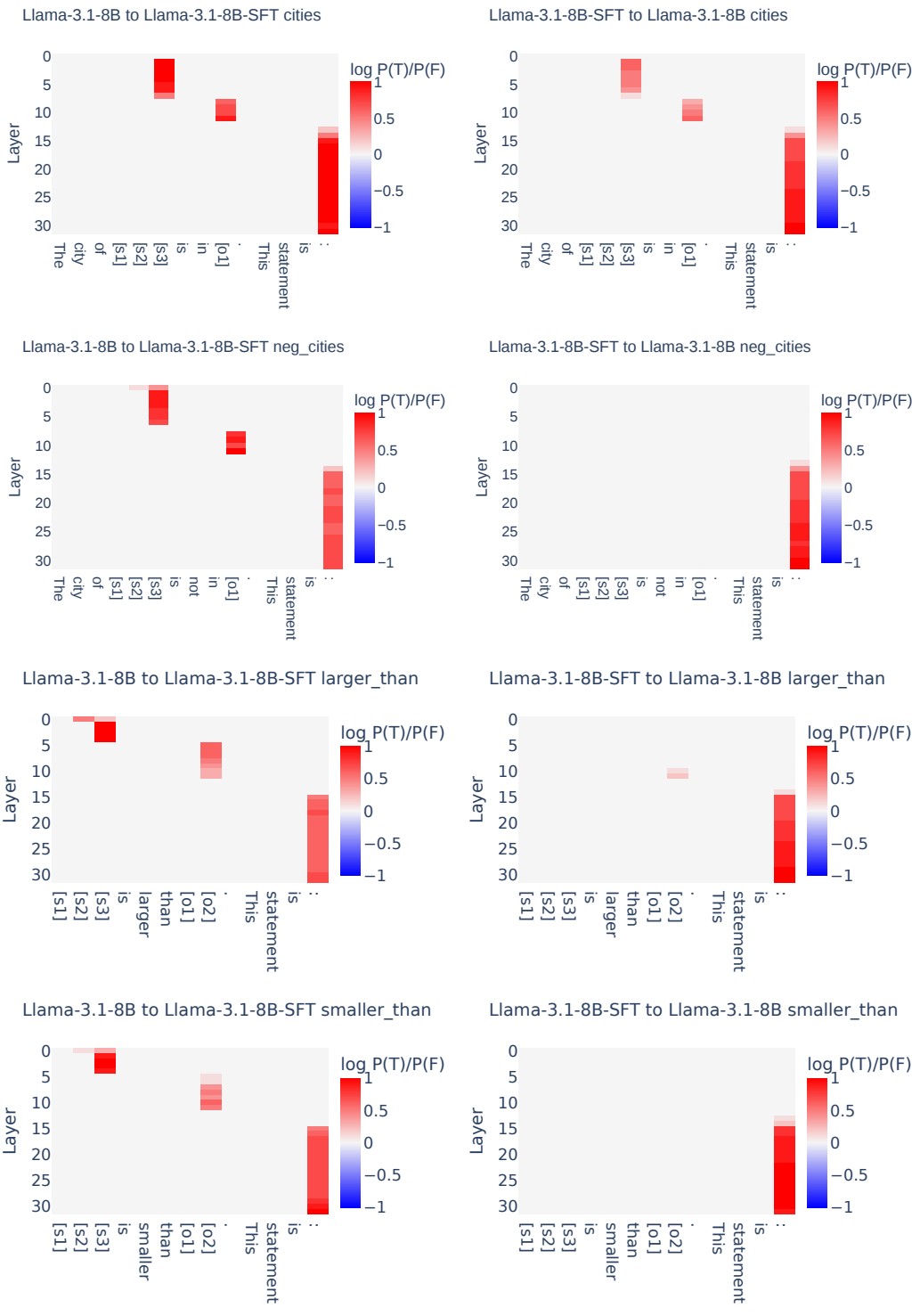

Figure 18: Cross-model patching results between llama-3.1-8b BASE and SFT.

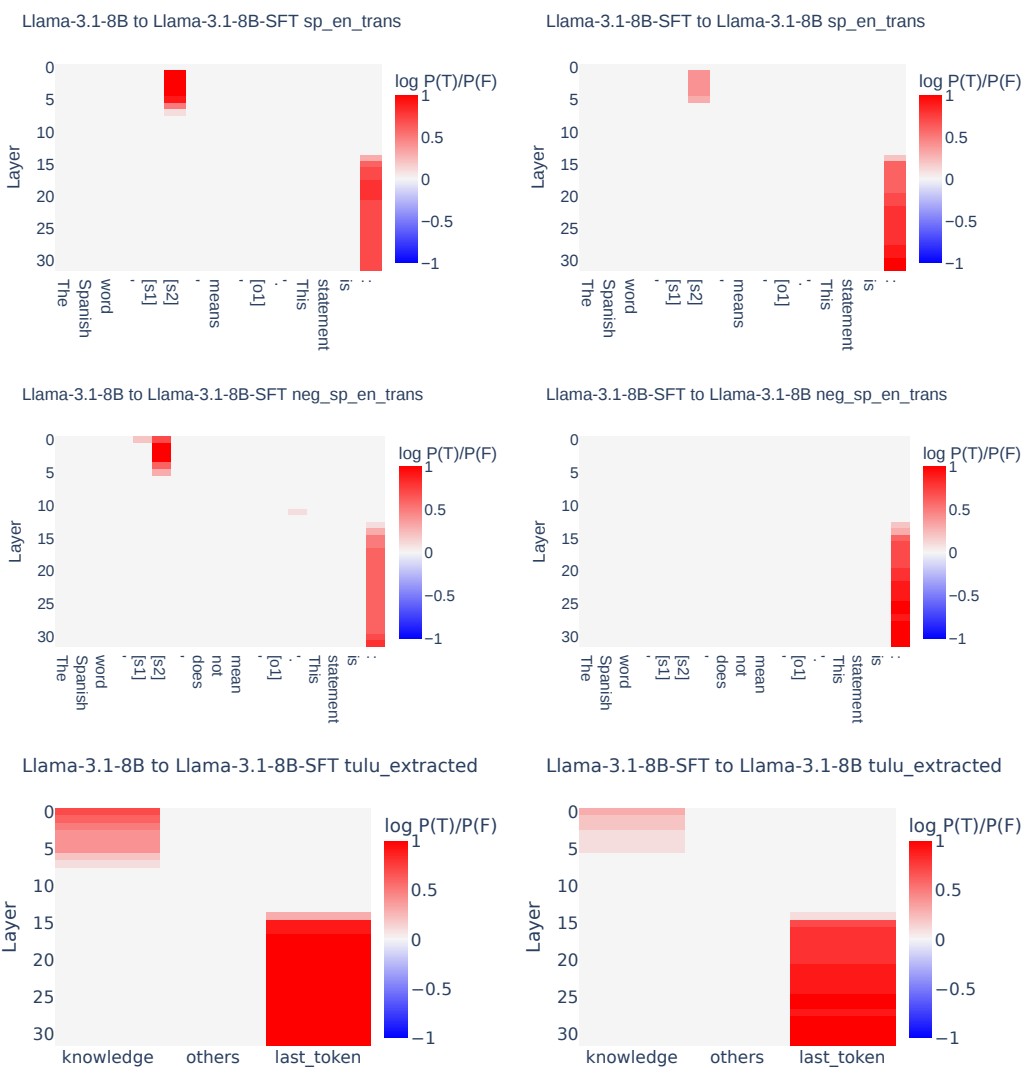

Figure 19: Cross-model patching results between llama-3.1-8b BASE and SFT (Continued).

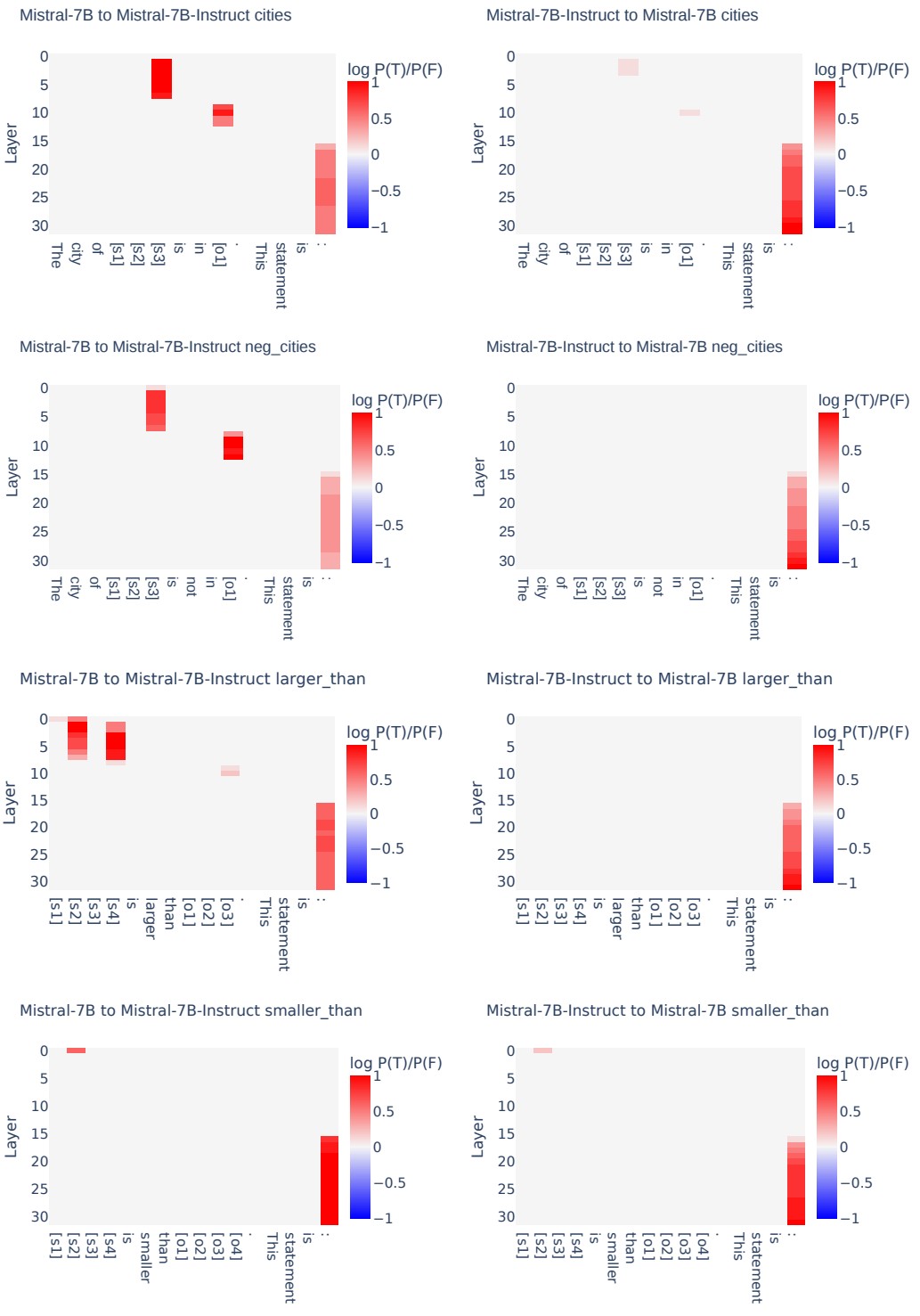

Figure 20: Cross-model patching results between Mistral-7B BASE and INSTRUCT.

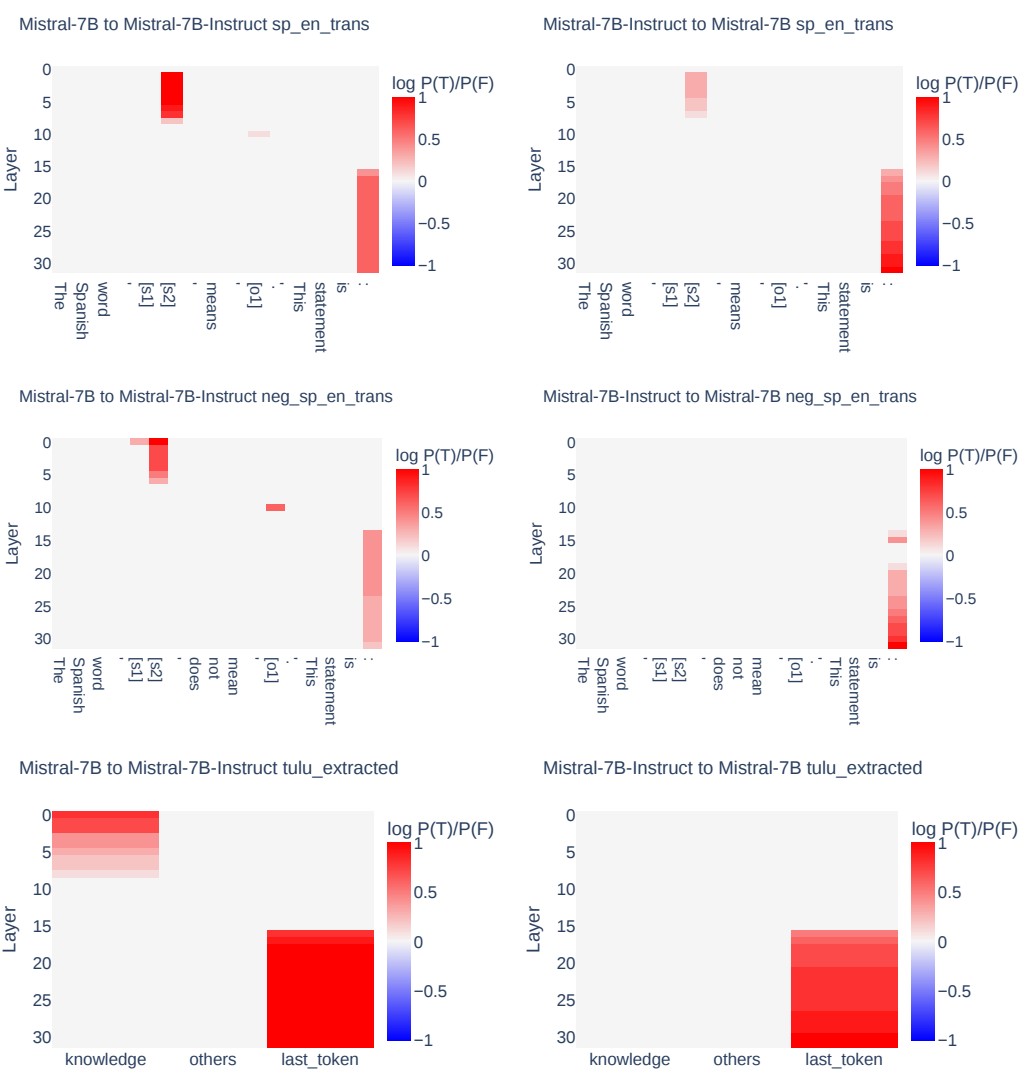

Figure 21: Cross-model patching results between Mistral-7B BASE and INSTRUCT (Continued).

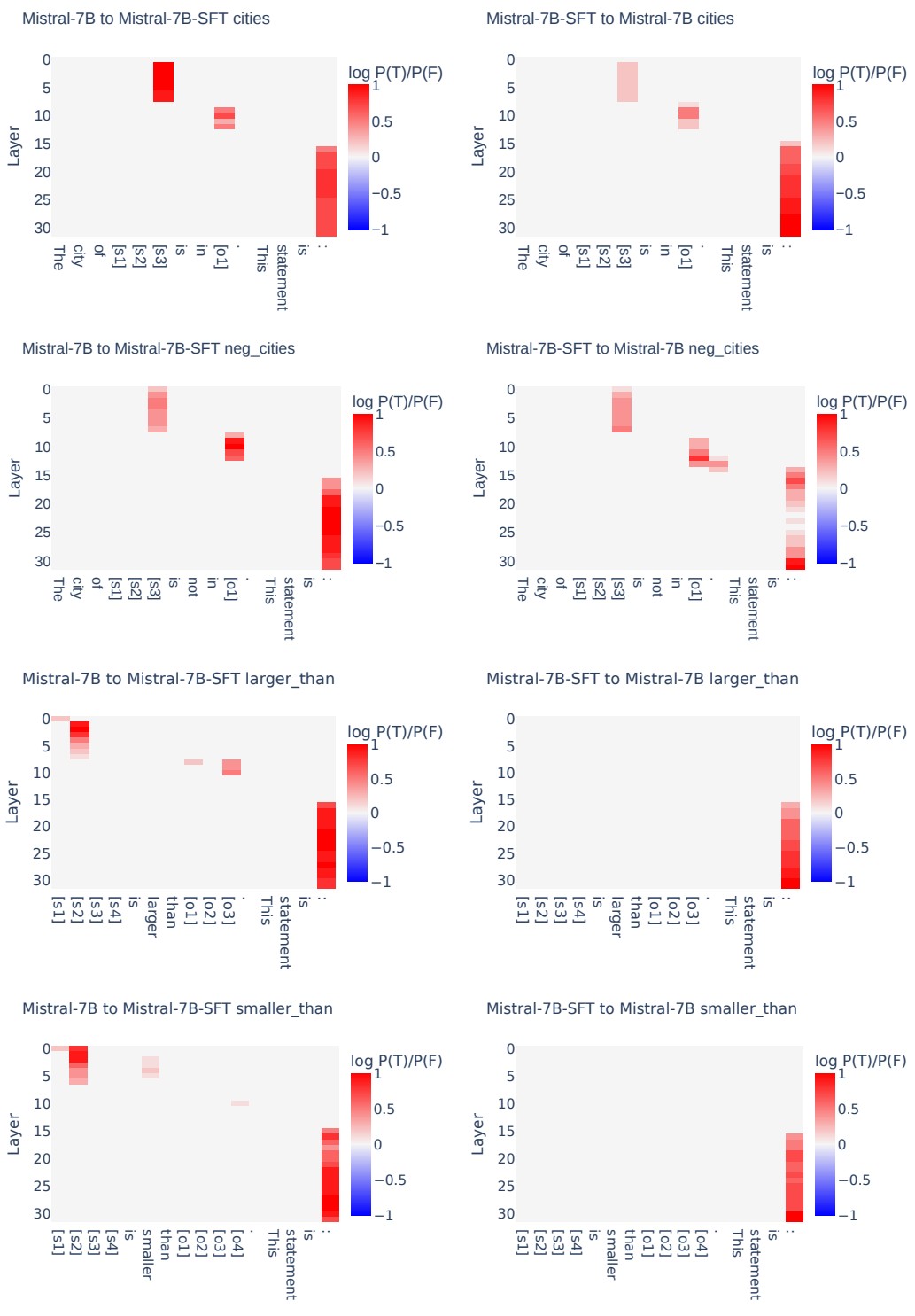

Figure 22: Cross-model patching results between Mistral-7B BASE and SFT.

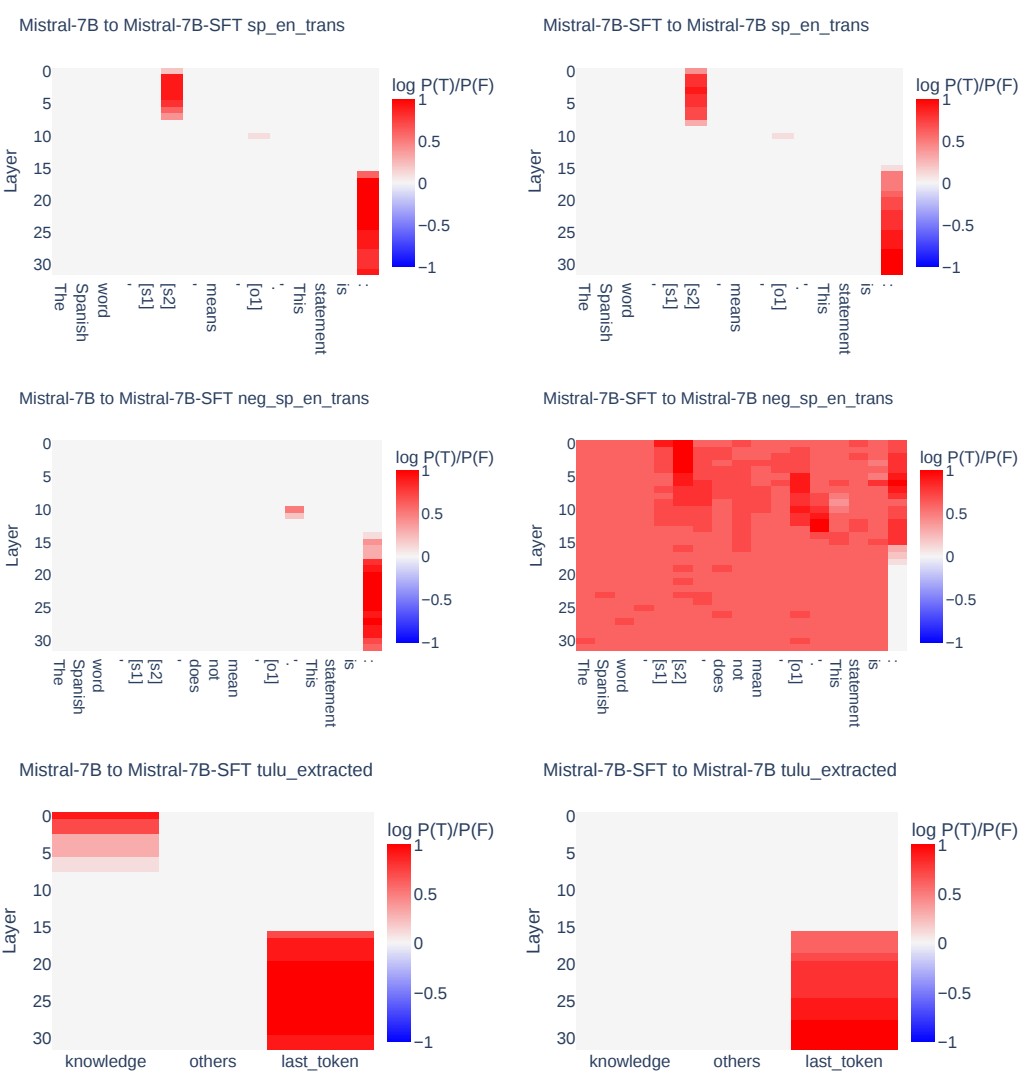

Figure 23: Cross-model patching results between Mistral-7B BASE and SFT (Continued).

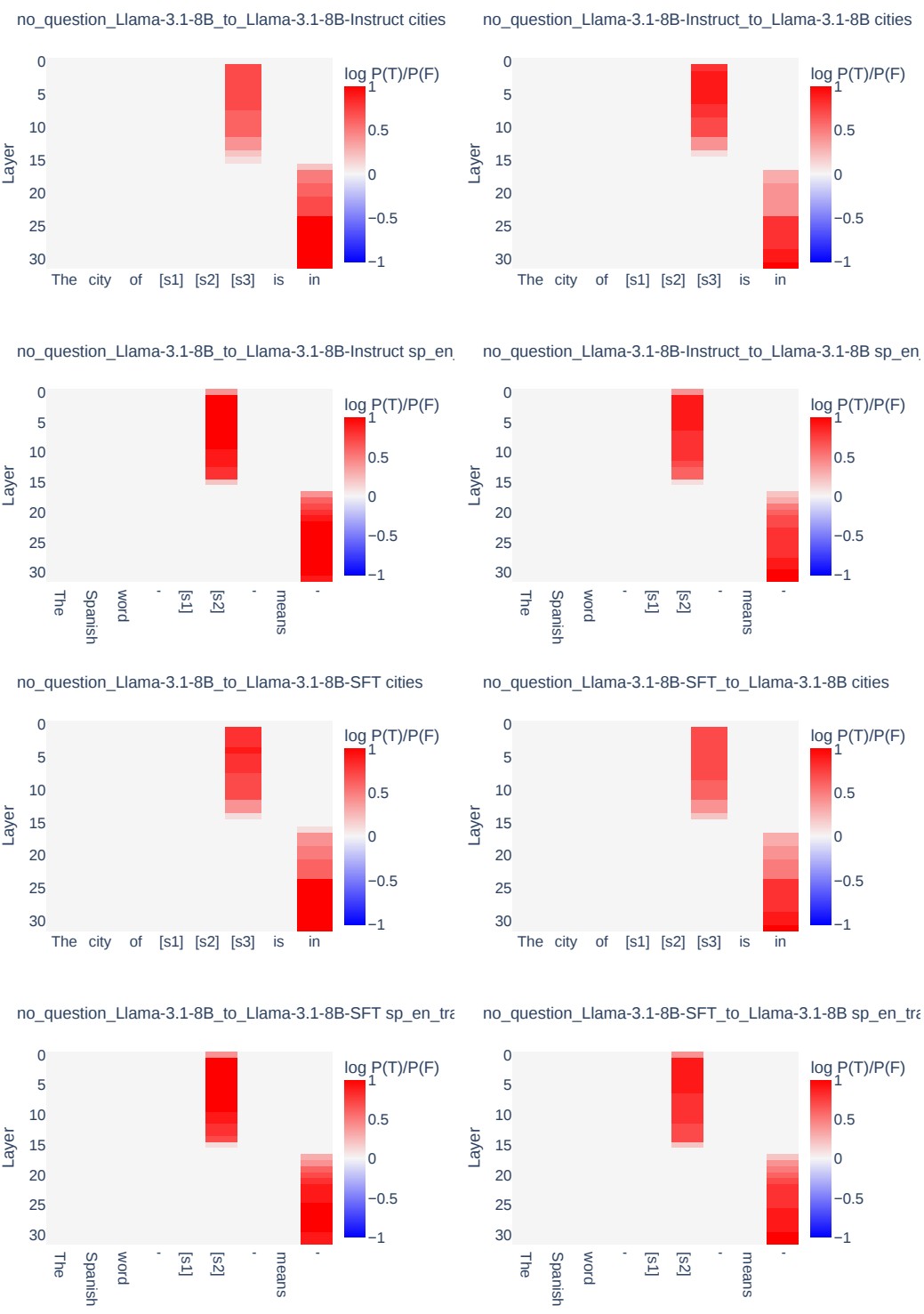

Figure 24: Cross-model patching results between Llama-3.1-8B BASE, INSTRUCT, and SFT in the traditional causal tracing setting.

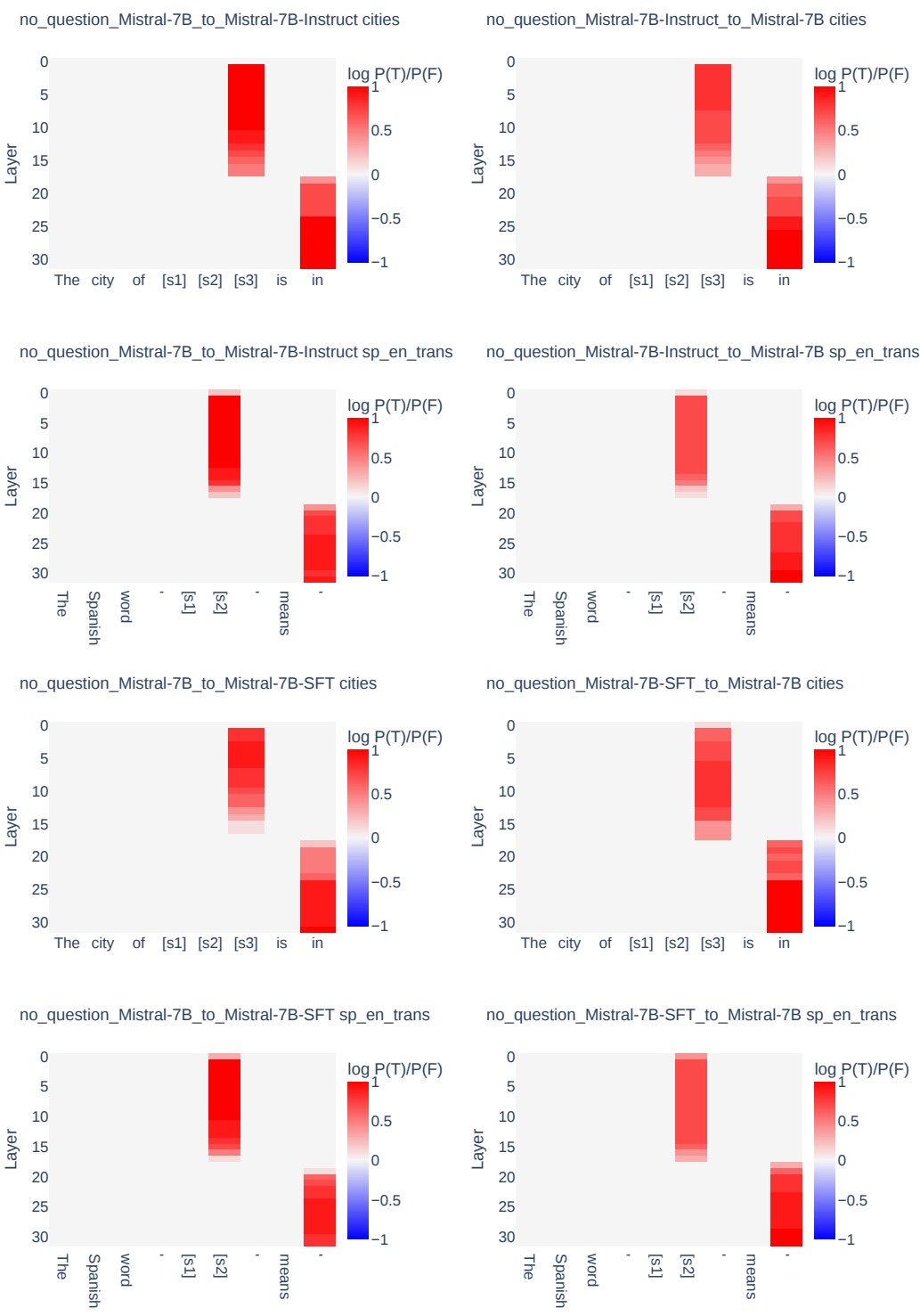

Figure 25: Cross-model patching results between Mistral-7B BASE, INSTRUCT, and SFT in the traditional causal tracing setting.

