# OpenReview forum: "How Post-Training Reshapes LLMs: A Mechanistic View on Knowledge, Truthfulness, Refusal, and Confidence"
_colmweb.org/COLM/2025/Conference — COLM 2025_

### Official Review · Reviewer_WfuA · 2025-05-03

**Rating:** 8
**Confidence:** 4
**Ethics Flag:** 1

**Summary:**

This paper investigates how post-training, including supervised fine-tuning (SFT) and instruction tuning, mechanistically alters LLMs. Instead of treating LLMs as black boxes, the authors conduct a systematic internal comparison of pre-trained (BASE) and post-trained (POST) models along four axes:
- Knowledge Storage and Representation: Post-training preserves knowledge locations but adapts and builds on prior knowledge representations.
- Truthfulness: Truthfulness can be captured via a linear direction in hidden space, which is largely shared between BASE and POST models.
- Refusal Behavior: Refusal directions exist but are less transferable between BASE and POST models.
- Confidence: Changes in model confidence cannot be explained by entropy neurons.

The findings are based on empirical analyses across multiple models and datasets and offer insights into which capabilities are pre-training artifacts versus those truly shaped by post-training. The results have implications for model editing, alignment, and interpretability.

**Questions To Authors:**

- Why is backward transfer (POST to BASE) possible for refusal but not forward (from BASE to POST)? Any hypotheses?
- Beyond entropy neurons, did you investigate other hidden state characteristics that might explain confidence differences?
- What models (sizes, architectures) and datasets were specifically used? Do you expect findings to generalize to much larger or more instruction-heavy models (e.g., GPT-4)?
- Did you explore other important axes such as reasoning, robustness, or chain-of-thought abilities? Why were these four perspectives chosen?

**Reasons To Accept:**

- This paper provides the mechanistic understanding of post-training effects, which is underexplored but highly relevant for future LLM development.
- The paper covers four complementary dimensions that collectively provide a deep mechanistic perspective, which is inspiring.
- The paper discovers that truthfulness can be transferred across BASE and POST is practically important for efficient model interventions.
- Insights in this paper are not just theoretical but suggest practical techniques for model steering, fine-tuning efficiency, and interpretability.

**Reasons To Reject:**

- While the results are empirical and insightful, some deeper theoretical grounding (e.g., a theoretical model supporting that linear directions capture truthfulness) is missing.
- The confidence section feels less robust compared to others; it largely concludes ``entropy neurons aren’t it’’ without exploring other possible factors.

---

> ### Author Response · Authors · 2025-06-03
>
> > Question #4: Did you explore other important axes such as reasoning, robustness, or chain-of-thought abilities? Why were these four perspectives chosen?
>
>
> We selected these four perspectives because we think that they represent fundamental aspects of language model behaviors, and they can be studied through well-defined experimental designs using available interpretability tools. While we acknowledge that other axes such as reasoning, robustness, and chain-of-thought abilities are also very important for understanding model behaviors and improving model performance, these areas present significant methodological challenges for existing interpretability tools. Specifically, reasoning and CoT abilities involve complex, multi-step processes that are difficult to isolate and measure in controlled, curated settings. We can analyze the model's output and performance as the reasoning ability, but it is difficult to analyze its internal mechanism. We view the development of more sophisticated interpretability tools capable of dissecting complex cognitive processes like reasoning as a promising direction for future research. We are also eager to find out how post-training reshapes the reasoning ability of LLMs.
>
>
> > Reasons to reject: While the results are empirical and insightful, some deeper theoretical grounding is missing. The confidence section feels less robust compared to others; it largely concludes ``entropy neurons aren’t it’’ without exploring other possible factors.
>
> Since LLMs are too complex, theoretical study is very difficult in this field. Nonetheless, there have been many useful interpretability works that give insights but not theory. The empirical results already provide good intuitions and could already benefit many applications, “could potentially lead to developing innovative ideas in this space”, as mentioned by Reviewer bJdd. For example, the intuitive understanding of the linear directions could benefit transfer-intervention by the linear direction of the Base model to the post-trained model or vice versa. We would be willing to extend our paper in the future if new theories could concretely explain these phenomena. For your concern about the entropy neurons, we have answered it in the questions above.
>
> Thank you again for your feedback!
>
> Reference:
>
> [1] Samuel Marks and Max Tegmark. 2024. The Geometry of Truth: Emergent Linear Structure in Large Language Model Representations of True/False Datasets.
>
> [2] Lennart Burger, Fred A. Hamprecht, and Boaz Nadler. 2024. Truth is universal: Robust detection of lies in llms.
>
> [3] Andy Zou, Zifan Wang, J. Zico Kolter, and Matt Fredrikson. 2023. Universal and transferable adversarial attacks on aligned language models.
>
> [4] Rohan Taori, Ishaan Gulrajani, Tianyi Zhang, Yann Dubois, Xuechen Li, Carlos Guestrin, Percy Liang, and Tatsunori B. Hashimoto. 2023. Stanford alpaca: An instruction-following llama model.
>
> [5] Andy Arditi, Oscar Obeso, Aaquib Syed, Daniel Paleka, Nina Panickssery, Wes Gurnee, and Neel Nanda. 2024. Refusal in language models is mediated by a single direction.
>
> [6] Alessandro Stolfo, Ben Wu, Wes Gurnee, Yonatan Belinkov, Xingyi Song, Mrinmaya Sachan, and Neel Nanda. 2024. Confidence regulation neurons in language models.
>
> [7] Rohan Taori, Ishaan Gulrajani, Tianyi Zhang, Yann Dubois, Xuechen Li, Carlos Guestrin, Percy Liang, and Tatsunori B. Hashimoto. 2023. Stanford alpaca: An instruction-following llama model.
>
> [8] Kaitlyn Zhou, Jena D. Hwang, Xiang Ren, Maarten Sap. 2024. Relying on the Unreliable: The Impact of Language Models’ Reluctance to Express Uncertainty.

---

> > ### Comment · Reviewer_WfuA · 2025-06-03
> > **Thanks for the Response**
> >
> > Dear Authors,
> >
> > Thank you very much for the clarification! Most of them have addressed my concerns.
> >
> > I appreciate the response, especially the newly conducted experiments. Including those in the revised draft will strengthen the paper.
> >
> > Overall, I acknowledge the quality of this work, and decide to keep my positive score.
> >
> > Best Regards,
> >
> > Reviewer WfuA

---

> > > ### Author Response · Authors · 2025-06-07
> > >
> > > Thank you again for your positive feedback! We truly appreciate your acknowledgment of our efforts. We will make sure to include all your suggested points in our updated paper, which will certainly increase our paper's quality.

---

> ### Author Response · Authors · 2025-06-03
>
> > Do you expect findings to generalize to much larger or more instruction-heavy models (e.g., GPT-4)?
>
> To verify whether our findings generalize to larger models, we conduct experiments on Llama-2-13B and Llama-2-13B-Instruct models. All our previous conclusions are consistently verified on this 13B model.
>
> For the knowledge storage perspective, we use the same experiment settings as our paper and use the metric as Table 1 to evaluate the causal tracing results. We conduct experiments on the *cities*, *sp_en_trans*, and *tulu_extracted* datasets. The results are shown below:
>
> |  | cities | sp_en_trans | tulu_extracted |
> | -------- | -------- | -------- | -------- |
> | Corr($M_{base}$, $M_{instruct}$) | 0.9885 | 0.9918 | 0.9970 |
> | $max\|M_{instruct} - M_{base}\|$ | 0.4 | 0.4 | 0.2 |
> | $max\|M_{instruct} - M_{base}\|_K$ | 0.4 | 0.4 | 0.1 |
>
> The causal tracing results of the based model and instruction-tuned model have a high correlation coefficient, and their maximum difference is low. It verifies our previous conclusion that post-training has little influence on knowledge storage locations.
>
>
> For the truthfulness probing experiments, we follow the same setting and metrics as Table 2, and the results are shown below:
>
>
> | Test Dataset | $p_{BASE} \to h_{BASE}$ | $p_{INS} \to h_{INS}$ / $p_{BASE} \to h_{INS}$ (Δ) |
> |--------------|-------------------------|--------------------------------------------------|
> | cities | 95.39 | 99.47 / 99.06 (-0.41) |
> | sp_en_trans | 96.89 | 96.33 / 90.68 (-5.65) |
> | inventors | 83.74 | 70.20 / 70.94 (+0.74) |
> | animal_class | 98.78 | 95.12 / 95.12 (+0) |
> | element_symb | 95.70 | 94.62 / 94.09 (-0.53) |
> | facts | 71.12 | 78.97 / 62.75 (-16.22) |
>
>
> For the truthfulness intervention experiments, we follow the same setting and metrics as Table 3, and the results are shown below:
>
> | Test Dataset | $t_{BASE} \to h_{BASE}$ | $t_{INS} \to h_{INS}$ / $t_{BASE} \to h_{INS}$ (Δ) |
> |--------------|-------------------------|--------------------------------------------------|
> | cities | 0.69 | 0.71 / 0.68 (-0.03) |
> | sp_en_trans | 0.83| 0.86 / 0.88 (+0.02) |
> | inventors | 0.66 | 0.64 / 0.67 (+0.03) |
> | animal_class | 0.72 | 0.73 / 0.74 (+0.01) |
> | element_symb | 0.79 | 0.84 / 0.83 (-0.01) |
> | facts | 0.68 | 0.63 / 0.66 (+0.03) |
>
> For the refusal intervention experiments, we follow the same setting and metrics as Table 4, and the results are shown below:
>
> | Test Dataset | baseline / $r_{BASE} \to h_{BASE}$ | baseline / $r_{INS} \to h_{INS}$ / $r_{BASE} \to h_{INS}$  |
> |--------------|-------------------------|--------------------------------------------------|
> | harmful | 0.24 / 0.37 | 0.99 / 0.59 / 0.99  |
> | harmless | 0.05 / 0.32 | 0.0 / 1.0 / 0.01 |
>
> These results confirm our previous conclusions that truthfulness directions are similar between the BASE and POST models but the refusal directions are different.
>
> For the entropy neuron experiments, all 10 top entropy neuron candidates are the same between the base model and instruction-tuned model. We also calculate the ratio of weight norm divided by the log LogitVar for the identified entropy neurons (as in Table 16 in Appendix E). The average ratio difference is only 0.003075 between the base model and the instruction-tuned model. This confirms our conclusion that the confidence difference between BASE and POST models cannot be attributed to entropy neurons.
>
> Unfortunately, we do not have the resources to conduct experiments on even larger models, but we expect our findings to generalize, and it remains a future work to verify them.

---

> ### Author Response · Authors · 2025-06-03
>
> Thank you for recognizing our impact, choosing four inspiring perspectives, and the practical importance! We would like to address your questions and concerns respectively as follows.
>
>
> > Question #1: Why is backward transfer (POST to BASE) possible for refusal but not forward (from BASE to POST)? Any hypotheses?
>
> We hypothesize that maybe the POST models develop a more sophisticated refusing mechanism with a more specified refusal direction. The acceptable refusal direction range in the POST model might be narrower than the BASE model, so the refusal direction in the BASE model does not work well in the POST models, but the refusal direction of the POST models might work well in the BASE model. It could be an interesting future direction to verify this hypothesis and dig deeper into the refusal behavior. This detailed analysis itself could become a separate paper. Our paper aims to provide a systematic analysis of the four perspectives that we think are important.
>
>
> > Question #2: Beyond entropy neurons, did you investigate other hidden state characteristics that might explain confidence differences?
>
> During our initial exploration, we considered developing an output-centered approach, where we rely on the model itself to express its confidence and try to attribute this confidence to model internal modules. However, we later found that letting the model itself determine if its output is confident or not is not reliable. As also pointed out by previous work [8], language models are reluctant to express uncertainties when answering questions even when they produce incorrect responses, and when explicitly prompted to express confidence, they tend to be overconfident. This is why we eventually chose the entropy neuron as our interpretability tool as it is intrinsic to the model.
>
> We acknowledge that the confidence section has some limitations. However, we think this finding is still valuable to report. Confidence calibration is an important part of understanding post-training's effects, and we identified entropy neurons as a potentially suitable tool to study this perspective based on prior works. Even though entropy neurons turned out not to be useful for this task, we believe reporting these results contributes valuable knowledge about what approaches don't work in this domain. It remains a future work to explore what leads to confidence change during post-training.
>
>
> > Question #3: What models (sizes, architectures) and datasets were specifically used?
>
> As explained in Section 3, we mainly conduct experiments on two representative LLM model families: Llama-3.1-8B and Mistral-7B-v0.3. For each of these two model families, we conduct experiments on the base model, instruction-tuned model, and SFT model. We use the base model and instruction-tuned model from the officially released model, and we use the SFT models from Llama-3.1-Tulu-3-8B-SFT and Mistral-7B-Base-SFT-Tulu2, which were finetuned on the Tulu dataset, since there was no official released SFT models of this size.
>
> For datasets, we choose the appropriate dataset for each analyzing perspective following previous papers, so that we can make sure our setting is reasonable and appropriate. For the knowledge and truthfulness perspectives, we use datasets from [1], where each dataset contains simple and unambiguous statements that are either true or false from diverse topics. For example, the *cities* dataset contains statements about cities and their countries, following the format “The city of [city] is in [country]”. To eliminate the concern that the datasets might be out-of-distribution for post-training, we curate a dataset that is in-distribution for SFT models. We curate the Tulu extracted dataset from the tulu-3-sft-mixture dataset that was used to finetune the Llama-3.1-8B-SFT model. We ensure every statement from the extracted dataset also appears in the tulu-v2-sft-mixture dataset that was used to finetune the Mistral-SFT model. For experiments on the refusal perspective, we follow [5] to use advbench [3] for harmful inputs and alpaca [7] for harmless inputs. The confidence perspective directly analyzes the entropy neurons, so we do not need any dataset. Details about our datasets are shown in Appendix A.

---

### Official Review · Reviewer_vQ6Y · 2025-05-12

**Rating:** 7
**Confidence:** 4
**Ethics Flag:** 1

**Summary:**

This paper systematically investigate how post-training method work on LLMs. It investigates from four critical perspectives: knowledge representation, truthfulness, refusal behavior and confidence. The paper reveal how post training shape LLMs internally.

**Questions To Authors:**

Do the larger models behave consistently across these four perspectives?

**Reasons To Accept:**

The paper reveals that it does not significantly alter knowledge storage locations or the direction of truthfulness. It adapts existing knowledge representations and develops new ones. Additionally, it changes the refusal direction and can modify confidence levels.
These findings may benefit various real-world applications.

**Reasons To Reject:**

The experiment limited the model size to under 8 billion parameters. Evaluating on larger models, such as those with 13 billion or even 40 billion parameters, could yield better results.

---

> ### Author Response · Authors · 2025-06-03
>
> Thank you very much for your positive feedback and recognition of our findings and potential benefits to real-world applications!
>
> > Evaluating on larger models, such as those with 13 billion or even 40 billion parameters, could yield better results. Do the larger models behave consistently across these four perspectives?
>
> To verify whether our findings generalize to larger models, we conduct experiments on Llama-2-13B and Llama-2-13B-Instruct models. All our previous conclusions are consistently verified on this 13B model.
>
> For the knowledge storage perspective, we use the same experiment settings as our paper and use the metric as Table 1 to evaluate the causal tracing results. We conduct experiments on the *cities*, *sp_en_trans*, and *tulu_extracted* datasets. The results are shown below:
>
> |  | cities | sp_en_trans | tulu_extracted |
> | -------- | -------- | -------- | -------- |
> | Corr($M_{base}$, $M_{instruct}$) | 0.9885 | 0.9918 | 0.9970 |
> | $max\|M_{instruct} - M_{base}\|$ | 0.4 | 0.4 | 0.2 |
> | $max\|M_{instruct} - M_{base}\|_K$ | 0.4 | 0.4 | 0.1 |
>
> The causal tracing results of the based model and instruction-tuned model have a high correlation coefficient, and their maximum difference is low. It verifies our previous conclusion that post-training has little influence on knowledge storage locations.
>
>
> For the truthfulness probing experiments, we follow the same setting and metrics as Table 2, and the results are shown below:
>
>
> | Test Dataset | $p_{BASE} \to h_{BASE}$ | $p_{INS} \to h_{INS}$ / $p_{BASE} \to h_{INS}$ (Δ) |
> |--------------|-------------------------|--------------------------------------------------|
> | cities | 95.39 | 99.47 / 99.06 (-0.41) |
> | sp_en_trans | 96.89 | 96.33 / 90.68 (-5.65) |
> | inventors | 83.74 | 70.20 / 70.94 (+0.74) |
> | animal_class | 98.78 | 95.12 / 95.12 (+0) |
> | element_symb | 95.70 | 94.62 / 94.09 (-0.53) |
> | facts | 71.12 | 78.97 / 62.75 (-16.22) |
>
>
> For the truthfulness intervention experiments, we follow the same setting and metrics as Table 3, and the results are shown below:
>
> | Test Dataset | $t_{BASE} \to h_{BASE}$ | $t_{INS} \to h_{INS}$ / $t_{BASE} \to h_{INS}$ (Δ) |
> |--------------|-------------------------|--------------------------------------------------|
> | cities | 0.69 | 0.71 / 0.68 (-0.03) |
> | sp_en_trans | 0.83| 0.86 / 0.88 (+0.02) |
> | inventors | 0.66 | 0.64 / 0.67 (+0.03) |
> | animal_class | 0.72 | 0.73 / 0.74 (+0.01) |
> | element_symb | 0.79 | 0.84 / 0.83 (-0.01) |
> | facts | 0.68 | 0.63 / 0.66 (+0.03) |
>
> For the refusal intervention experiments, we follow the same setting and metrics as Table 4, and the results are shown below:
>
> | Test Dataset | baseline / $r_{BASE} \to h_{BASE}$ | baseline / $r_{INS} \to h_{INS}$ / $r_{BASE} \to h_{INS}$  |
> |--------------|-------------------------|--------------------------------------------------|
> | harmful | 0.24 / 0.37 | 0.99 / 0.59 / 0.99  |
> | harmless | 0.05 / 0.32 | 0.0 / 1.0 / 0.01 |
>
> These results confirm our previous conclusions that truthfulness directions are similar between the BASE and POST models but the refusal directions are different.
>
> For the entropy neuron experiments, all 10 top entropy neuron candidates are the same between the base model and instruction-tuned model. We also calculate the ratio of weight norm divided by the log LogitVar for the identified entropy neurons (as in Table 16 in Appendix E). The average ratio difference is only 0.003075 between the base model and the instruction-tuned model. This confirms our conclusion that the confidence difference between BASE and POST models cannot be attributed to entropy neurons.
>
> Unfortunately, we do not have the resources to conduct experiments on even larger models, but we expect our findings to generalize, and it remains a future work to verify them.

---

> > ### Comment · Reviewer_vQ6Y · 2025-06-11
> > **Thanks for your response**
> >
> > Dear authors,
> >
> > Most of my concerns are well addressed, so I keep my positive score.
> >
> > Best wishes,
> > Reviewer vQ6Y

---

> > > ### Author Response · Authors · 2025-06-11
> > >
> > > Thank you again for your recognition! We truly appreciate your positive feedback, and we will make sure to include all your suggested points in our updated paper.

---

### Official Review · Reviewer_K28J · 2025-05-13

**Rating:** 6
**Confidence:** 4
**Ethics Flag:** 1

**Summary:**

This paper presents a mechanistic evaluation of how post-training has changed the pre-trained language models in terms of knowledge, truthfulness, refusal, and confidence. The paper finds: (1) post-training does not change the locations where factual knowledge is stored, but changes the knowledge representations; (2) Post-training does not change the truthfulness representations by much; (3) Post-training changes the refusal representations; (4) Entropy neurons, commonly used in the literature to explain the confidence calibration of LLMs, cannot explain the difference between confidence of the post-trained model and the pre-trained model.

**Questions To Authors:**

1. Line 152: How do you ensure that "every statement from tulu_extracted also appears in the tulu dataset"? Do you look for the verbatim occurrence of the statement?

2. Truthful interventions: Line 261 shows that the scaling factor for truthful intervention can only be +1 or -1. In prior works on intervening and steering models for truthfulness (e.g. [5]), it seems that the choice of scaling factor has a significant impact on the intervention results. Is there a reason for choosing this pair of scalars? Would the conclusions be different if the scalars are tuned on a validation set of better intervention effects?

**Reasons To Accept:**

1. The paper provides a holistic evaluation of the differences between post-trained models and pre-trained models across a wide range of behaviors that people generally care about post-training.

2. The paper conducts comprehensive experiments on various models with well-accepted mechanistic interpretability methods, enabling it to draw general conclusions about post-training.

3. The paper has very solid experimental designs for many settings: For example, the paper takes the distribution shift after post-training on knowledge-intensive prompts into account in Section 4, and this is usually overlooked by prior works.

4. The paper draws interesting conclusions about how post-training changes the truthfulness and confidence behaviors and could be potentially useful for future research in LLM alignment.

**Reasons To Reject:**

1. The experimental setup of the knowledge analyses raises some concerns. While the knowledge analyses aims at determining the knowledge storage and the knowledge representation have been changed after post-training, the paper uses causal tracing to extract the knowledge location and representation by the change of log probability between "This statement is TRUE" and "This statement is FALSE" (Equation (1)). This seems to me more like extracting the model's understanding of the truthfulness of a given statement rather than extracting the knowledge. There is a difference between knowing if a given statement is true or false (truthfulness, or the capability to validate) and knowing the right answer to make up a true statement (knowledge, or the capability to generate), and it is known in the literature that LLMs are inconsistent for these two capabilities (also called generator-validator consistency as in [2]). Moreover, Section 4 of the paper follows the setup of [1] while [1] uses that setup for probing truthfulness rather than knowledge. Thus, I have a concern that the experimental setup in the paper probes a different phenomenon from knowledge. I think it would be more appropriate to follow another practice in prior works on mechanistic analyses of knowledge (e.g. [3]): measuring the patching's influence by the change in the probability of the entity of interest (e.g. the object of the knowledge triple).

2. The refusal analyses do not add new insights from what's known in the literature. Although the paper conducts very thorough experiments on refusal, the exact conclusion it draws has already been discussed in [1] (in Appendix J). The only difference is that [1] compares the cosine similarity of the refusal directions extracted from chat and base models while the paper uses causal tracing.

3. The methods used for analyses are not well-motivated. The paper does not use a consistent set of analytical methods, and does not provide a clear motivation for choosing one mechanistic interpretability analysis method over another: it uses causal tracing for analyzing knowledge and refusal, linear probe (and causal tracing, which is used to build the probe) for truthfulness, and entropy neuron for confidence. It is unclear to me why the same probing analyses are not used for refusal since the paper mentions that refusal can also be represented in a linear direction, similarly to truthfulness, and also, why causal tracing and probing are not used for analyzing confidence calibration. For the latter, I understood prior works have found that entropy neurons can be used to explain confidence calibration. However, it is noteworthy that prior works have found that there might exist knowledge neurons [4] and truthful attention head [5] to explain knowledge and truthfulness respectively.

4. This paper could be substantially strengthened if it could provide deeper analyses on one of the four phenomena of interest, especially analyses of how different post-training methods have different effects. For example, Table 4 already shows some differences in transferability of the refusal direction between BASE -> SFT and BASE -> INS, potentially showing an impact of SFT vs. RLHF for refusal.

Despite these, I still think the paper has solid experiments and gives new insights about truthfulness and confidence calibration. I am happy to adjust my scores if the authors can address my concerns.

[1] Samuel Marks and Max Tegmark. 2024. The Geometry of Truth: Emergent Linear Structure in Large Language Model Representations of True/False Datasets.

[2] Xiang Lisa Li, Vaishnavi Shrivastava, Siyan Li, Tatsunori Hashimoto and Percy Liang. 2024. “Benchmarking and Improving Generator-Validator Consistency of Language Models.

[3] Kevin Meng, David Bau, Alex J Andonian, Yonatan Belinkov. 2022. Locating and Editing Factual Associations in GPT.

[4] Damai Dai, Li Dong, Yaru Hao, Zhifang Sui, Baobao Chang, Furu Wei. 2022. Knowledge Neurons in Pretrained Transformers.

[5] Kenneth Li, Oam Patel, Fernanda Viégas, Hanspeter Pfister, Martin Wattenberg. 2023. Inference-Time Intervention: Eliciting Truthful Answers from a Language Model

---

> ### Author Response · Authors · 2025-06-03
>
> > Question 2: Is there a reason for choosing this pair of scalars? Would the conclusions be different if the scalars are tuned on a validation set of better intervention effects?
>
> We are following the original scalar setup in [1], where the scaling factors are constrained to +1 or -1. However, we acknowledge that prior works on model steering for truthfulness [5] have demonstrated that scalar choice can impact intervention effectiveness. To directly address this concern and ensure our conclusions are robust across different scaling configurations, we conducted additional experiments varying the scaling factor from 1 to 10 with a step size of 1 on the llama-3.1-8b and llama-3.1-8b-instruct model pair.
>
> For each dataset and model combination, we identified the optimal scaling factor that maximized the performance. We report the optimal scaling factors in the table below. INS→INS means using the truthfulness direction of the instruction-tuned model to intervene in the instruction-tuned model. BASE->INS means using the truthfulness direction of the base model to intervene in the instruction-tuned model. Coef records the optimal scaling factor, and IE (intervention effect) is our metric evaluating the intervention as in Table 3. Delta is the difference between BASE→INS IE and INS→INS IE.
>
> | Val Dataset       | INS→INS IE | INS→INS Coef | BASE→INS IE | BASE→INS Coef | Delta  |
> |---------------|-----------|-------------|-----------|-------------|--------|
> | cities        | 0.8880    | 2.00        | 0.8968    | 1.00        | 0.0088 |
> | sp_en_trans   | 0.8484    | 2.00        | 0.8409    | 3.00        | -0.0075|
> | inventors     | 0.7063    | 1.00        | 0.7192    | 1.00        | 0.0129 |
> | animal_class  | 0.7973    | 2.00        | 0.8298    | 1.00        | 0.0325 |
> | element_symb  | 0.7582    | 2.00        | 0.7697    | 1.00        | 0.0115 |
> | facts         | 0.6185    | 1.00        | 0.6560    | 1.00        | 0.0375 |
>
> The results show that base direction and instruct direction have similar intervention performance when both use their own best-performing scalar choice. Although the intervention results are a little sensitive to the scaling factors, our main conclusions remain valid even when scaling factors are tuned: both the base direction and instruct direction can be used to intervene on the instruct model with equal effectiveness.
>
>
> Reference:
>
> [1] Samuel Marks and Max Tegmark. 2024. The Geometry of Truth: Emergent Linear Structure in Large Language Model Representations of True/False Datasets.
>
> [2] Xiang Lisa Li, Vaishnavi Shrivastava, Siyan Li, Tatsunori Hashimoto and Percy Liang. 2024. “Benchmarking and Improving Generator-Validator Consistency of Language Models.
>
> [3] Kevin Meng, David Bau, Alex J Andonian, Yonatan Belinkov. 2022. Locating and Editing Factual Associations in GPT.
>
> [4] Damai Dai, Li Dong, Yaru Hao, Zhifang Sui, Baobao Chang, Furu Wei. 2022. Knowledge Neurons in Pretrained Transformers.
>
> [5] Kenneth Li, Oam Patel, Fernanda Viégas, Hanspeter Pfister, Martin Wattenberg. 2023. Inference-Time Intervention: Eliciting Truthful Answers from a Language Model
>
> [6] Andy Arditi, Oscar Obeso, Aaquib Syed, Daniel Paleka, Nina Panickssery, Wes Gurnee, and Neel Nanda. 2024. Refusal in language models is mediated by a single direction.
>
> [7] Andrew Lee, Xiaoyan Bai, Itamar Pres, Martin Wattenberg, Jonathan K. Kummerfeld, and Rada Mihalcea. 2024. A mechanistic understanding of alignment algorithms: A case study on dpo and toxicity.

---

> > ### Comment · Reviewer_K28J · 2025-06-06
> >
> > Thanks to the authors for such comprehensive responses! I really appreciate the efforts the authors put into the rebuttal. I have updated my scores accordingly.
> >
> > > Additional results of using the ROME setting; Additional results of using the optimal scaling factor
> >
> > Thank you so much for running additional experiments! The results make a lot of sense to me. I think including these results in the final version of the paper will definitely strengthen the argument made in the paper.
> >
> > > Motivations for using the specific interpretability methods for each setting
> >
> > Thank you for including the detailed explanations of the motivations for using the specific methods for each setting in your paper. Although I do not totally agree with all of the motivations the authors provided in the explanations (for example, I do not agree that probing is inappropriate for understanding confidence calibration because of the difficulty of separating confident and non-confident input-output pairs; there exist datasets such as ConflictingQA [1] and AmbigQA [2] for which such pairs can be reasonably constructed), I think including these in the beginning of each section to explain the selected methodology would make the paper better motivated.
> >
> > [1] Alexander Wan, Eric Wallace, Dan Klein. What Evidence Do Language Models Find Convincing?
> >
> > [2] Sewon Min, Julian Michael, Hannaneh Hajishirzi, Luke Zettlemoyer. AmbigQA: Answering Ambiguous Open-domain Questions

---

> > > ### Author Response · Authors · 2025-06-10
> > >
> > > Thank you very much for your recognition and the increased score! Also thank you very much for your additional feedback!
> > >
> > > We appreciate you bringing ConflictingQA[1] and AmbigQA[2] to our attention. ConflictingQA creates controversial questions with conflicting web evidence to examine what types of evidence the LLM finds convincing. AmbigQA addresses ambiguous open-domain questions by requiring the LLM to identify multiple plausible answers and generate disambiguated questions for each answer.
> > >
> > > However, these two datasets do not directly satisfy our needs for studying model confidence. There can be different definitions of "confidence". The "confidence" we study is the model uncertainty about its **output**. Specifically, it means the probability distribution the target LLM assigned to the output tokens, where concentration means more confident, and uniform means less confident (we will revise our paper to further clarify this point). ConflictingQA assigns confidence labels to input evidence using an **external model** (FLAN-large) and studies the correlation between convincingness and evidence confidence. This approach measures uncertainty of the **input** evidence but does not directly reflect confidence of the target LLM as in our definition. AmbigQA, on the other hand, focuses on ambiguous questions that require clarification, which primarily examines LLM's robustness and disambiguation capabilities rather than its confidence.
> > >
> > > We do see potentials for using these datasets to study LLM's confidence, but specifically doing probing for confidence is still challenging. A successful linear probing depends on the assumption that the target concept can be represented linearly as a direction in hidden states. This assumption has been verified for truthfulness [3] and refusal [4] by previous papers, so we did linear probing for them. However, to the best of our knowledge, none of the previous papers has shown that confidence can be represented cleanly as a direction in LLM's hidden states. Non-linear probing might be possible, but such complex probing model can lose the benefits and insights of interpretability.
> > >
> > > Nevertheless, your recommended papers offer valuable insights, and we believe they could inform the future development of better tools and datasets for studying model confidence. We hope our explanation has clarified the concerns. We are happy to dicuss any remaining concerns and would appreciate score adjustment if all concerns are cleared.
> > >
> > > References:
> > >
> > > [1] Alexander Wan, Eric Wallace, Dan Klein. What Evidence Do Language Models Find Convincing?
> > >
> > > [2] Sewon Min, Julian Michael, Hannaneh Hajishirzi, Luke Zettlemoyer. AmbigQA: Answering Ambiguous Open-domain Questions.
> > >
> > > [3] Samuel Marks and Max Tegmark. The Geometry of Truth: Emergent Linear Structure in Large Language Model Representations of True/False Datasets.
> > >
> > > [4] Andy Arditi, Oscar Obeso, Aaquib Syed, Daniel Paleka, Nina Panickssery, Wes Gurnee, and Neel Nanda. Refusal in language models is mediated by a single direction.

---

> ### Author Response · Authors · 2025-06-03
>
> > Prior works have found that there might exist knowledge neurons (KN) [4] and truthful attention head [5] to explain knowledge and truthfulness, respectively.
>
> As we discussed above, our rule of thumb is to use the latest and most suitable methods for each analysis.
>
> Knowledge Neurons (KN), as a great method for knowledge analysis, does not fit our purpose as good as causal tracing. KN was originally proposed as a method on **encoder-decoder models**, specifically BERT, whereas causal tracing was more recently proposed as a method on **decoder-only models** explored in this work, e.g., Llama, Mistral, etc.
>
> Moreover, even though KN has been adapted for decoder-only models in subsequent work, empirical evidence from [3] demonstrates that causal tracing outperforms KN during model editing tasks, indicating that causal tracing provides more precise localization of stored knowledge. Given that transfer knowledge editing represents a potential application of our findings, this superior performance makes causal tracing more suitable for our knowledge analysis objectives. Additionally, since our truthfulness analysis follows [1] in employing causal tracing for intervention layer selection, adopting causal tracing for knowledge analysis ensures methodological consistency across our experimental framework.
>
> We chose not to use truthful attention heads as it is also not the most suitable method for our analysis.
>
> First, our experimental design follows [1], which focuses on datasets of "clear, simple, and unambiguous factual statements" to isolate truthfulness mechanisms without confounding factors. This approach differs from [5], which employs datasets such as TruthfulQA that contain intentionally misleading questions designed to test models' susceptibility to common misconceptions—a valuable feature for evaluating robustness but not aligned with our specific experimental objectives.
>
> Second, truthful attention heads have primarily been validated on adversarial or ambiguous datasets, but their effectiveness on our curated setting of straightforward factual statements remains unproven. Since we are adhering to the controlled experimental framework established in [1], we opted for the MM-probe method, which has demonstrated reliable performance in similar settings with clear factual statements.
>
>
>
> > Reason to reject #4: This paper could be substantially strengthened if it could provide deeper analyses on one of the four phenomena of interest, especially analyses of how different post-training methods have different effects.
>
> We agree with you that doing experiments with different post-training algorithms is very meaningful, while analyzing their effects can be a separate paper for each perspective. For example, paper [7], which we cited in related work, explores how DPO affects the toxicity of the models. Our work aims to provide a systematic analysis of all four perspectives that we believe are important for understanding the post-training effects.
>
> > Question 1: Line 152: How do you ensure that "every statement from tulu_extracted also appears in the tulu dataset"? Do you look for the verbatim occurrence of the statement?
>
> There are two Tulu datasets: tulu-3-sft-mixture dataset, which was used to finetune Llama-3.1-8B-SFT, and tulu-v2-sft-mixture, which was used to finetune the Mistral-SFT model. flan_v2_converted is a common subset of tulu-3 and tulu-v2. We use GPT-4o to extract many factual knowledge statements from flan_v2_converted. We manually picked 100 factual knowledge statements from the generated statements that are high-quality, valid factual knowledge, and indeed appear in the flan_v2_converted dataset. Therefore, these factual knowledge statements must exist in the fine-tuning dataset of Llama-3.1-8B-SFT and Mistral-SFT models. While we reformatted the syntax to maintain consistency with other datasets, the underlying content and knowledge remain intact. We have recorded the indices mapping each tulu_extracted datapoint to its location in the tulu-3-sft-mixture dataset. To ensure full transparency and reproducibility, we will include these mapping indices in our upcoming code release.

---

> ### Author Response · Authors · 2025-06-03
>
> > Reason to reject #2: The refusal analyses do not add new insights from what's known in the literature.
>
>
> We guess you might be referring to paper [6] rather than paper [1]. The key distinction between our paper and [6] lies in our experimental methodology: In Appendix J of paper [6], the authors computed cosine similarities between **activations** (from base and chat models) and a **single refusal direction** (extracted from the chat model). Our approach directly computes cosine similarity between the **base model's refusal direction** and the **chat model's refusal direction**. Additionally, we conduct **transfer intervention** experiments to causally validate our findings. In conclusion, while we acknowledge the contribution of paper [6], our own experiment settings are different from it and do provide additional insights into refusal directions in BASE and POST models.
>
>
>
>
>
>
> > Reason to reject #3: The methods used for analyses are not well-motivated. The paper does not use a consistent set of analytical methods.
>
>
> We totally agree with you that there are often several different interpretability methods available for analyzing each type of model behavior, and the differences, pros, and cons between them can be subtle. Our rule of thumb is to use the latest and most suitable methods for each of the four perspectives.
>
> We would like to first clarify our analysis techniques. We use causal tracing for analyzing knowledge; probing and intervention (and causal tracing, which is used to select the layer to build the probe) for truthfulness; intervention for refusal; and entropy neurons for confidence. We discuss the subparts one by one in the following.
>
> > It is unclear to me why the same probing analyses are not used for refusal
>
> We made such a design because the difference between truthfulness and refusal is subtle. The probing analysis for truthfulness aims to understand if the model can internally distinguish between true and false **inputs**. However, refusal is a property centered at **outputs**. The refusal interpretability method we explored in this paper is the best method we know, and it learns the refusal direction from harmful/harmless data. So the learned direction represents an "internal belief of harmfulness", which corresponds to how the model distinguishes a statement being **harmful** vs **harmless**, but not necessarily whether the model **refuses** to answer an input or not. Although they are related, they are not exactly the same. To avoid potential mislead and to directly test refusal behavior, we use the Refusal Score, which is intervention-based and allows us to directly evaluate the refusal results based on outputs. We see this as a more reasonable evaluation design that can be more easily extended to future methods for learning better refusal directions.
>
>
> In fact, in paper [6]'s Appendix J, their cosine similarity experiment can be considered as a form of direct probing on the input activations (the only difference between cosine similarity and direct probing is that linear probing means a sigmoid function on top of the dot product of the probing direction and input activations). However, due to the gap we discussed above, we believe the evaluation based on Refusal Score is more aligned with what we need.
>
>
>
> > why causal tracing and probing are not used for analyzing confidence calibration.
>
> Unlike binary knowledge verification tasks, where factual statements can be cleanly categorized as true or false, model confidence exists on a continuous spectrum that cannot be easily categorized into 'confident' and 'not confident' categories. This fundamental difference makes the causal tracing approach unsuitable, as it requires distinct clean and corrupt runs to isolate causal effects. Similarly, differences in mean probing methods depend on having well-defined input pairs representing confident versus non-confident states, which is problematic when confidence is inherently graded rather than categorical. To the best of our knowledge, none of the previous papers have utilized causal tracing or probing to analyze confidence calibration.
>
> We therefore chose to focus on entropy neurons based on established evidence from prior work demonstrating that these neurons function as internal confidence calibrators within the model's architecture. This approach allows us to study confidence mechanisms without artificially imposing binary distinctions on what is naturally a continuous phenomenon.

---

> ### Author Response · Authors · 2025-06-03
>
> Thank you for your detailed and thoughtful feedback and recognition of our holistic evaluation, comprehensive experiments, conclusions, and potential usefulness! We would like to address your concerns and questions respectively as follows.
>
> > Reason to reject #1: The experimental setup of the knowledge analyses raises some concerns, and why we did not follow the ROME setting.
>
> We chose this setup from [1] because of the following considerations. First, our setting (e.g., "The city of Toronto is in Canada. This statement is:") can detect knowledge storage in both subject and object. In contrast, the ROME setting [3] provides the subject and lets the model outputs the object, e.g., "The city of Toronto is in". It can only detect knowledge storage in the subject. Second, our setting can test a wider range of factual knowledge. ROME setting evaluates patching's influence to the output logit of the correct objct, so it must have a single correct object, such as the country of a city. We did causal tracing experiments on 7 datasets, but only 2 of them do have a single correct object. They are:
>
> * cities: The statement is "The city of xxx is in xxx", where the object should be the country of the city.
> * sp_en_trans: The statement is "The Spanish word 'xxx' means 'xxx'", where the object should be the English translation (although it might have some synonyms, it is almost fixed).
>
> The other 5 datasets do not have a single correct object. They are:
>
> * neg_cities: The statement is "The city of xxx is not in xxx", where the object can be any country except for the correct one.
> * larger_than: The statement is "xxx is larger than xxx", where the object can be any number smaller than the subject.
> * smaller_than: The statement is "xxx is smaller than xxx", where the object can be any number larger than the subject.
> * neg_sp_en-trans: The statement is "The Spanish word 'xxx' does not mean 'xxx'", where the object can be any English word except for the correct translation.
> * tulu_extracted: The statements cover a wide range of factual knowledge, such as "The digits of Pi are infinite." and "A car is something you drive, not cook with."
>
> Third, these datasets have “clear scope” and “are simple, uncontroversial, and unambiguous” [1]. So we expect LLMs' output should align well with their knowledge.
>
> Nonetheless, we fully understand your concerns based on [2], which is very reasonable. Following your suggestions, we conduct experiments based on ROME setting [3] on *cities* and *sp_en_trans* datasets. In our previous experiments, a pair of statements contain a true statement and a false statement, where they have the same object but different subjects. Now we directly let the model output the object. We denote the model's output object for one statement as $O_1$ and the output for another statement as $O_2$. We use $log \frac{P(O_1)}{P(O_2)}$ to denote whether the patching is successful. We use the same metrics as Table 1 to evaluate the results. The results of Llama-3.1-8B model family are shown below:
>
> |  | cities | sp_en_trans |
> | -------- | -------- | -------- |
> | Corr($M_{base}$, $M_{instruct}$) | 0.9961 | 0.9982 |
> | $max\|M_{instruct} - M_{base}\|$ | 0.1 | 0.1 |
> | $max\|M_{instruct} - M_{base}\|_K$ | 0.1 | 0.1 |
> | Corr($M_{base}$, $M_{SFT}$) | 0.9968 | 0.9989 |
> | $max\|M_{SFT} - M_{base}\|$ | 0.1 | 0.1 |
> | $max\|M_{SFT} - M_{base}\|_K$ | 0.1 | 0.1 |
>
> The results of Mistral-7B model family are shown below:
>
> |  | cities | sp_en_trans |
> | -------- | -------- | -------- |
> | Corr($M_{base}$, $M_{instruct}$) | 0.9982 | 0.9981 |
> | $max\|M_{instruct} - M_{base}\|$ | 0.1 | 0.1 |
> | $max\|M_{instruct} - M_{base}\|_K$ | 0.1 | 0.1 |
> | Corr($M_{base}$, $M_{SFT}$) | 0.9900 | 0.9959 |
> | $max\|M_{SFT} - M_{base}\|$ | 0.3 | 0.3 |
> | $max\|M_{SFT} - M_{base}\|_K$ | 0.3 | 0.3 |
>
> The results are even stronger than Table 1. It verifies our conclusion that post-training has little influence on knowledge storage locations. The results on cross-model patching is also simialr to our previous conclusions, though being slightly different. The backward transfer (patching from post-trained model to base model) is slightly available, but not as good as forward transfer (patching from base model to post-trained model). We are not able to upload visualization results in the rebuttal. We will update our paper to show all the new visualization results.

---

### Official Review · Reviewer_bJdd · 2025-05-13

**Rating:** 9
**Confidence:** 4
**Ethics Flag:** 1

**Summary:**

This paper provides an in-depth mechanistic analysis on BASE (pretrained), SFT (Instruction fine-tuned) and INSTRUCT (post RLHF) models using the method proposed by Marks and Tegmark (2024). The paper analyzes the models across four perspectives - (i) understanding "storage" of facts, (ii) truthfulness direction and transfer across model training stages (iii) similar analysis on refusal direction and (iv) differences in confidence. The paper provides various insights potentially useful for model understanding.

**Reasons To Accept:**

- The paper presents very interesting and important mechanistic analysis of models across various training stages. This is timely and can have a significant impact in advancing the interpretability literature for understanding models across training stages for fact understanding

- The analysis are thorough, they have performed it across various models to validate findings, showing generalizable findings.

- The tasks that the authors analyzed is also very impactful, understanding truthfulness and refusal directions in various training stages could potentially lead to developing innovative ideas in this space

- The paper identifies that while truthfulness shows high cosine similarity, refusal doesn't. There is scope for another paper in this topic alone.

**Reasons To Reject:**

- Nothing major. But section 7 is rather not well-written. While explained briefly, the paper could use explaining how entropy neurons were applied in their analysis in a bit more detail.

---

> ### Author Response · Authors · 2025-06-03
>
> Thank you for recognizing our paper as timely and impactful, thorough and generalizable, and mechanistically insightful!
>
> > Section 7 is rather not well-written. While explained briefly, the paper could use explaining how entropy neurons were applied in their analysis in a bit more detail.
>
> We agree that the writing of Section 7 can be further improved. The current discussion of entropy neurons is insufficient, mainly due to space limitations. In our updated version, we will include the following discussion about our methodology of applying entropy neurons:
>
> Entropy neurons are identified by considering weight norm and logit attribution. First, we compute the logit attribution for each neuron in the final MLP layer by projecting their output weights onto the vocabulary space through the unembedding matrix, as shown in Equation 4 of our paper. This projection approximates each neuron's direct effect on the final prediction logits (its effect on each token's output logit). We then calculate the variance of this normalized projection (LogitVar), where low LogitVar indicates a relatively balanced contribution across all vocabulary tokens rather than promoting specific tokens. Entropy neurons typically have a large weight norm (so that they are influential) but low LogitVar (so that they have a relatively balanced contribution across all vocabulary tokens). Our identification process first selects the top 25% of neurons with the largest weight norms, and from this subset, we identify the 10 neurons with the lowest LogitVar values as entropy neurons. This methodology follows established practices from prior work [1,2] and captures neurons that modulate output entropy without significantly affecting token ranking.
>
> In our analysis comparing BASE and POST models, we found substantial overlap in identified entropy neurons, with a highly similar ratio of weight norm divided by log LogitVar (Table 16 in Appendix E). It suggests that the confidence regulation mechanism of entropy neurons remains largely unchanged during post-training. This finding indicates that the observed confidence calibration differences between BASE and POST models likely stem from more subtle mechanistic changes that require sophisticated interpretability tools beyond current entropy neuron analysis to fully understand.
>
> We will carefully check the whole paper to ensure each part is clear.
>
> [1] Wes Gurnee, Theo Horsley, Zifan Carl Guo, Tara Rezaei Kheirkhah, Qinyi Sun, Will Hathaway, Neel Nanda, and Dimitris Bertsimas. 2024. Universal neurons in gpt2 language models.
>
> [2] Alessandro Stolfo, Ben Wu, Wes Gurnee, Yonatan Belinkov, Xingyi Song, Mrinmaya Sachan, and Neel Nanda. 2024. Confidence regulation neurons in language models.

---

### Decision · Program_Chairs · 2025-07-08

**Decision:**

Accept

**Comment:**

This paper explores the effects of post-training on the internal mechanisms of LLMs from four dimensions.  Knowledge storage locations are shown to be similar before and after post-training, although the analysis is limited in scope.  Internal representations of truthfulness are similar; similar vectors can intervene on truthfulness before and after training. Similar effects are shown for refusal. Finally, analysis of model confidence shows that this changes substantially between base and post-trained models, but there is no easy explanation.

The reviewers are generally positive about this paper. It is overall well-written and has some interesting experiments. However, I do think there are some valid criticisms by reviewer K28J. In general, I think the paper studies several different phenomena incompletely.  I can imagine a stronger paper that takes one of these effects and really analyzes it in a lot more depth, finding something more fundamental as a result. As it is, the experiments differ between settings and don't feel strongly unified except insofar as they focus on models before and after post-training.